# A Boosting-Driven Model for Updatable Learned Indexes

## Abstract

Learned Indexes (LIs) represent a paradigm shift from traditional index structures by employing machine learning models to approximate the Cumulative Distribution Function (CDF) of sorted data. While LIs achieve remarkable efficiency for static datasets, their performance degrades under dynamic updates: maintaining the CDF invariant ($\sum F(k) = 1$) requires global model retraining, which blocks queries and limits the Queries-per-second (QPS) metric. Current approaches fail to address these retraining costs effectively, rendering them unsuitable for real-world workloads with frequent updates. In this paper, we present a Sigmoid-based model, *an efficient and adaptive learned index that minimizes retraining cost* through three key techniques: *(1) A Sigmoid boosting approximation* technique that dynamically adjusts the index model by approximating update-induced shifts in data distribution with localized sigmoid functions that preserves the model's $\epsilon$-bounded error guarantees while deferring full retraining. *(2) Proactive update training* using Gaussian mix models (GMMs) that identify high-update-probability regions for strategic placeholder allocation that speeds up updates coming in these slots. *(3) A neural joint optimization framework* that continuously refining both the sigmoid ensemble and GMM parameters via gradient-based learning. We rigorously evaluate our model against state-of-the-art updatable LIs on real-world and synthetic workloads, and show that it *reduces retraining cost by* $20\times$ *while showing up to* $3\times$ *higher QPS and* $1000\times$ *lower memory usage.*

## 1 Introduction

**Context.** Learned Indexes (LIs) (Ding et al., 2020; Chatterjee et al., 2024; Li et al., 2020; Heidari et al., 2025b; Tang & et al., 2020; Kipf et al., 2020; Heidari & Zhang, 2025; Kim et al., 2024; Lan et al., 2024; Heidari et al., 2025a) represent a paradigm shift in database indexing by replacing traditional pointer-based structures (e.g., B-trees) with machine learning models that directly approximate the *cumulative distribution function (CDF)* of sorted data. At their core, LIs treat the indexing problem as a modeling task: given a sorted dataset, they learn a mapping of keys to their positions by fitting the CDF $F(k)$, which describes the probability that a key $\leq k$ exists in the dataset. This approach enables *single-step position predictions* during queries, bypassing the $O(\log n)$ traversals of B-trees (Ferragina & Vinciguerra, 2020a).

The Recursive Model Index (RMI) (Kraska et al., 2018) exemplifies this approach through a hierarchical model ensemble, where higher levels narrow the search range and the leaf models predict exact positions. Practical implementations like ALEX (Ding et al., 2020), DobLIX (Heidari et al., 2025b), LISA (Li et al., 2020) optimize this further using *piecewise linear regression*, partitioning the key space into segments modeled by: pos = $a \times k + b \pm E$, where $a, b$ are learned parameters and $E$ bounds the prediction error, ensuring correctness via a final localized search ($\epsilon$-bounded error), and achieving 2–10$\times$ faster lookups than B-trees for static data (Ferragina & Vinciguerra, 2020a).

However, a fundamental limitation of LIs stems from their inherent assumption of static data distributions. Since the CDF maps keys to global ranks and index must compensate for the CDF shift to maintain monotonicity, a single insertion at $u$ shifts the target rank of all subsequent keys by exactly 1 and necessitates non-local adjustments. This creates a 'step function' change in the distribution that cannot be corrected by simple global scaling. Consequently, maintaining a precise CDF typically requires expensive model retraining. This requirement makes it particularly challenging to preserve model accuracy in dynamic workloads (Sabek & Kraska, 2023).

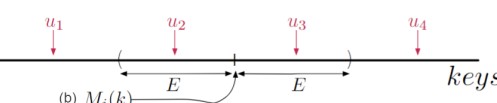

Figure 1: **(a) Retrain cost on three updatable LIs. Number of retrain occurrences and average retrain duration are shown by bar plots and $\times$ markers, respectively. (b) Impact of update on an LI model** $M_i(.)$

Previous studies (Ge & et al., 2023; Zhang et al., 2024; Ding et al., 2020; Galakatos et al., 2019; Liang et al., 2024; Tang & et al., 2020; Ferragina & Vinciguerra, 2020b; Wu & Ives, 2024; Heidari et al., 2025b) attempt to address this problem. Aside from methods such as DobLIX (Heidari et al., 2025b), which implement LI in read-only structures and thus avoid update issues, other approaches have limitations because they ignore the training cost of LI models (Wongkham et al., 2022; Ge et al., 2023; Heidari et al., 2025a). Figure 1(a) quantifies the retraining overhead of three state-of-the-art updatable LIs. The results reveal two key insights: (1) LIPP (Wu et al., 2021) and DILI (Li et al., 2023) exhibit frequent retraining (approximately once every 500 updates), and (2) while ALEX (Ding et al., 2020) shows fewer retraining events, each retraining incurs significantly higher latency. These excessive retraining overheads render current LI systems impractical for update-intensive workloads, common in real-world applications, as each retraining operation blocks query processing and severely degrades system QPS.

**Motivation.** Figure 1(b) illustrates how a single update affects the key space of an LI model. This model, denoted as $M_i$ [1], with a non-zero error $E$, maps a range across a key space. Incoming updates, viewed as a random variable, impact this range in four distinct ways: ($u_1$) shifts all elements uniformly without changing the range size; ($u_2$) expands the range by shifting $M_i(k)$ to the right; ($u_3$) enlarges the range on the right side; ($u_4$) does not alter the range size.

Each update induces a step-wise displacement in the model's prediction space. We reformulate retraining as a distributional prediction problem, where sigmoid functions approximate these discrete shifts smoothly. The differentiable properties of the sigmoid enable gradual adaptation of the model, deferring full retraining of CDF. For a single update, the sigmoid $\sigma_0(k, A, \omega_u, u)$ will be added to the model $M_i(k)$. When incremental updates exceed the capacity of a single sigmoid, we introduce a *SigmaSigmoid* ensemble to capture cumulative effects: $M_i'(k) = M_i(k) + \sum_{i=1}^{\mathcal{N}} \sigma(k, A_i, \omega_i, \phi_i)$, where $\mathcal{N}$ dynamically grows with new update patterns. This boosting approach amortizes retraining costs by (1) preserving existing model parameters and (2) isolating adjustments to affected regions via localized sigmoid terms (See Appendix C.1 for a detailed motivational example).

**Approach.** We propose **Sig**ma-**Sig**moid **Model**ing, (**Sig2Model**), *an efficient updatable learned index that minimizes retraining through adaptive sigmoid approximation and proactive workload modeling.* Our approach introduces three key techniques: *First*, Sig2Model employs a *sigmoid boosting technique* that dynamically adjusts the index model to incoming updates. By approximating the step-wise patterns of updates with localized sigmoid functions, each acting as a weak learner, the system incrementally corrects model errors while maintaining $\epsilon$-bounded error guarantee. This allows continuous model adaptation without immediate retraining. *Second*, we develop a *Gaussian Mixture Model (GMM)* component that predicts update patterns, enabling strategic insertion of placeholders in high-probability update regions. This anticipatory mechanism significantly reduces future retraining needs by pre-allocating space in frequently modified index segments for future updates. *Third*, Sig2Model integrates these components through a *unified neural architecture* that jointly optimizes both the sigmoid ensemble parameters and GMM distributions via gradient-based learning. The system processes updates in batched operations through a dedicated buffer, with a control module monitoring model error bounds to trigger retraining only when necessary. During retraining phases, the system simultaneously refines both the sigmoid approximations and placeholder allocations based on observed update patterns.

Sig2Model adapts an LI model[2] to support efficient updates using the above three techniques. The end-to-end workflow of Sig2Model for update and lookup operations is shown in Figure 2. For updates, the system first checks whether the incoming update $u$ exists in the current index domain $D^\tau$.

---

[1]The full table of notations is provided in Appendix A.

[2]Our implementation builds upon RadixSpline (Kipf et al., 2020).

Figure 2: **Sig2Model Overview.**

If a pre-allocated placeholder is available, $u$ is inserted directly. Otherwise, it is staged in the Update Buffer. This buffer serves as an efficient batching mechanism, accumulating updates until a threshold ($\rho$) is reached, at which point neural network training is triggered to optimize the SigmaSigmoid and GMM parameters. When the number of active sigmoids reach system capacity $\mathcal{N}$ during this process, a full retraining is initiated. For lookups, queries first probe the buffer for key $k$. On a miss, the inference module applies the learned SigmaSigmoid adjustments to the base model to perform a final search within the LI's error bound $\pm E$. If $k$ is not found in the $E$ range, Sig2Model performs an exhaustive search and triggers the retraining signal.

**Contributions.** The main contributions of this paper are as follows: *(1) Index Model Approximation (Π)*: We propose Sig2Model, a novel method that leverages sigmoid functions as weak approximators, similar to boosting techniques, to dynamically update the LI model. This approach significantly reduces the need for retraining, offering an efficient and adaptive solution. *(2) Probabilistic Update Workload Prediction (Ψ)*: We use GMM in Sig2Model to predict high-density regions in the key distribution, allowing strategic placeholder placement and postponing retraining. *(3) Neural Joint Optimization*: We propose two neural networks architecture ($NN_\Pi$, $NN_\Psi$) connected via a shared layer ($NN_c$) that continuously fine-tunes both $\Pi$ and $\Psi$ in a background process. *(4) Comprehensive Experimental Evaluation*: We rigorously evaluate the performance of Sig2Model through extensive experiments, and show over $20\times$ reduce in retraining time and an increase of up to $3\times$ in QPS compared to the state-of-the-art LI solutions.

## 2   INDEX MODEL APPROXIMATION

In this section, we present the **SigmaSigmoid Boosting** approach for capturing model updates efficiently. We assume that updates originate from an unknown distribution, denoted as $\mathcal{D}_{update}$. The bias introduced by updates, which can be modeled as a step function, is approximated using smooth, differentiable sigmoid functions (see Motivation in Section 1). This approximation avoids abrupt changes and delays retraining by gradually adjusting the model.

For a model $M$ at stage $\tau$, if dataset $D^\tau$ (with $s_\tau$ keys) receives an update $u$ between keys $k_j$ and $k_{j+1}$, the index model can be adjusted without full retraining using $D^\tau \cup \{u\}$. If a single sigmoid fails to capture an update, additional sigmoids can be introduced. The combined effect of these sigmoids, termed the SigmaSigmoid function, adjusts the model and postpones retraining.

The resulting SigmaSigmoid-based model, $M'_l(k, \Pi)$, is defined as:

$$M'_l(k, \Pi) = M_l(k) + \underbrace{\sum_{i=1}^{\mathcal{N}} \sigma(k, A_i, \omega_i, \phi_i)}_{\text{Adjustments to capture updates}} \tag{1}$$

$\Pi = \{(A_i, \omega_i, \phi_i)\}_{i \in \mathcal{N}}$ parameterizes $\mathcal{N}$ sigmoids, where $A_i$ controls amplitude, $\omega_i$ controls step steepness, and $\phi_i$ centers the sigmoid around update locations. The system capacity $\mathcal{N}$ may differ from the number of updates $|U|$, often with $|U| \geq \mathcal{N}$. This model raises theoretical questions and defines system limits, discussed in Appendix G. Ideally, $round(M_l(k_{j+1})) = round(M_l(k_j)) + 1$, requiring a new model $M'_l$ such that:

$$1 \leq \left| M'(k_{j+1}, \Pi) - M'(k_j, \Pi) \right| \leq 2. \tag{2}$$

It is important to note that Equation 1 theoretically admits a trivial solution for $|U| \leq \mathcal{N}$ (by setting $\phi_i = u_i, A_i = 1$). While this guarantees that the optimization landscape contains reachable global minima for small batches, our training objective (Equation 3) is designed to generalize beyond this trivial regime to handle the target scenario where $|U| \gg \mathcal{N}$.

Figure 3: **Neural Networks Architectures. The multi-layer neural network $NN_C$(c) processes sequential data batches for continuous learning. It employs highway networks, ReLU activations, and dropout layers to extract patterns and prevent overfitting. To address catastrophic forgetting** (Kemker et al., 2018)**, two strategies are used:** *Replay Buffer* (Di-Castro et al., 2022) **and** *Elastic Weight Consolidation* (Kirkpatrick et al., 2017)**. The network outputs are fed into two subnets,** $NN_\Pi$**(a) and** $NN_\Psi$**(b), each containing specific tasks. We explain** $NN_C$ **architecture in Appendix** D**.**

## 2.1 Optimization Objectives

Given a dataset $D$ and update set $U$ (collected from buffer $B$ of size $\rho$), we adjust each key's index based on updates $x \in U$ where $x < k$. The updated model $M^\star(\cdot)$ is obtained by retraining on $D \cup U$. Our goal is to find parameters $\Pi$ for $M'(\cdot, \Pi)$ that minimize:

$$\mathcal{L}(\Pi) = \frac{1}{|D \cup U|} \sum_{k \in D \cup U} \left| M'(k, \Pi) - M^\star(k) \right| + \frac{\gamma}{N} f(|\Pi|) \tag{3}$$

where $f(\cdot)$ is monotonically increasing, and $|\Pi| \leq \mathcal{N}$. The optimization problem is:

$$\arg \min_{M'(\cdot, \Pi) \in \mathbb{M}} \mathcal{L}(\Pi)$$
$$\text{s.t.} \quad \mathbb{E}_{u \sim \mathcal{D}_{update}} \left[ |M'(u, \Pi) - M^\star(u)| \right] \leq \alpha, \tag{4}$$
$$\mathbb{P}\left[ |M'(k, \Pi) - M'(u, \Pi)| < E_\Pi \right] \leq \beta \quad \forall u \sim \mathcal{D}_{update}, k \in D \cup U.$$

Solving Equation 4 is challenging for arbitrary hypothesis spaces $\mathbb{M}$. We use sigmoids as base models, transforming the original index function $M$. In this equation, $\alpha$ bounds prediction bias, and $\beta$ specifies the allowable error level. When $\beta = 0$, the problem becomes NP-Hard, requiring a complete search in $\mathbb{M}$ or even infeasible. $E_\Pi$ present the measure of confusion in LI models' prediction can be viewed as the variance of the index estimator for incoming $u \sim \mathcal{D}_{update}$. An unbiased estimator ($\alpha = 0$) is preferred over high variance, as variance only expands the last-mile search range. In addition, the target key in the lower variance has a higher likelihood of presence in the center of the predicted range. While optimization minimizes the error, the $\epsilon$-bound is strictly enforced during inference: if a key is not found within the predicted range, the system falls back to an exhaustive search and triggers a retraining signal, ensuring correctness is never violated.

To ensure a non-empty feasible solution space for the optimization problem described in Equation 4, it is necessary to show that maintaining unbiasedness ($\alpha \approx 0$) leads to increased variance, thereby bias-variance analysis indicates that $E_\Pi \propto \frac{1}{\alpha}$. Additionally, the second constraint in Equation 4 accounts for the most probable incoming updates to optimize intervals effectively, and is the relaxed derived as a modification of Equation 2, as shown in Theorem 1 (proof in Appendeix F):

**Theorem 1** *If $\mathcal{D}_{update}$ is bounded, Equation 2 satisfies $\mathbb{P}\left[ |M'(k, \Pi) - M'(u, \Pi)| < E_\Pi \right] \leq \beta$ for all $u \sim \mathcal{D}_{update}, k \in D \cup U$.*

## 2.2 Neural Unit for Fine-Tuning

The parameter set $\Pi = \{(A_i, \omega_i, \phi_i)\}_{i \in \mathcal{N}}$, where $\mathcal{N}$ is a hyperparameter, requires dynamic fine-tuning to adjust the LI model $M$. We implement this through a neural network $NN_\Pi$ with two fully-connected input networks (having $\mathcal{N}$ and $3\mathcal{N}$ parameters respectively) connected to a shared layer $NN_C$ as shown in Figure 2.

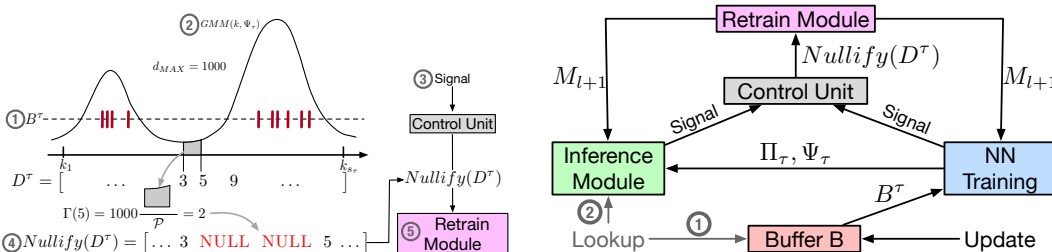

Figure 4: **(a) Steps for using the estimated update workload to produce placeholders. (b) Retrain Policy**

To minimize last-mile search errors, $NN_\Pi$ is optimized to near-overfit conditions using mean squared error (MSE) as the loss function: $MSE(X^\tau, Y^\tau) = \frac{1}{|X|} \sum_{(X_i^\tau, Y_i^\tau) \in (X^\tau, Y^\tau)} (\hat{I}_i - Y_i)^2$, where $\hat{I}_i$ is computed via Equation 1.

The cost function incorporates a regularization term to minimize sigmoid usage for new buffer entries $B^\tau$:

$$\mathcal{L}_\Pi(X^\tau, Y^\tau) = MSE(X^\tau, Y^\tau) + \frac{\gamma}{\mathcal{N}} \sum_{j=1}^{\mathcal{N}} \frac{\rho A_j}{d} \exp\left(1 - \frac{A_j}{d}\right) \tag{5}$$

where $\rho$ is the buffer size, $d$ is the normalization constant, and $\gamma$ is an experimentally-tuned hyperparameter.

From Equation 3, we derive $|\Pi| \approx \sum \mathbb{I}_{(A,.,.) \in \Pi, A \neq 0}$ and $f(x) = \frac{x}{d} \exp(1 - \frac{x}{d})$ (which is monotonic). This regularization design preferentially penalizes smaller amplitudes ($A_j$), pushing them toward zero while allowing larger amplitudes to cover more examples, consistent with the monotonicity of the index function. The optimization process solves Equation 4 using the prepared data from Section 4.1.

### 2.3 LOOKUP AND UPDATE OPERATIONS

**Lookup.** For a query key $k$, Sig2Model performs: (1) Check buffer $B^\tau$ for $k$ (O(log $\rho$)); (2) If not found, compute adjusted prediction: $M_l'(k_{emb}, \Pi^\tau) = M_l(k_{emb}) + \sum_{(A,\omega,\phi) \in \Pi^\tau} \sigma(k_{emb}, A, \omega, \phi)$; (3) Search positions $[M_l'(k_{emb}) - E, M_l'(k_{emb}) + E]$ in $D^\tau$; (4) If not found, trigger exhaustive search and retrain signal.

**Update.** For an incoming update $u$: (1) Check if position $M_l'(u_{emb}, \Pi^\tau)$ has a pre-allocated placeholder; (2) If yes, insert directly (O(1)); (3) Otherwise, add $u$ to buffer $B^\tau$ (O(log $\rho$)); (4) When $|B^\tau| = \rho$, trigger neural network training (Algorithm 1); (5) If $|\Pi| = \mathcal{N}$, trigger full retrain.

## 3 UPDATE WORKLOAD TRAINING

In this section, we introduce Sig2Model's **update workload training**, which is called *Nullifier* component. Sig2Model trains a probabilistic model on the incoming update distribution, and whenever a retraining signal is raised, it uses this trained distribution to put the update placeholder to improve system performance.

Nullifer manages data with a workload $\mathcal{D}_{keys}$ by creating space for updates based on their distribution. It operates on input data $D^\tau = [k_1, k_2, \ldots, k_{s_\tau}]$ drawn from $\mathcal{D}_{keys}$, with a maximum gap $d_{MAX}$, and extends $D^\tau$ using the update distribution $\mathcal{D}_{update}$ (see Figure 2(a)). The gap size between keys $k_i$ and $k_j$ ($i < j$) is calculated as

$$GS(k_i, k_j) = \left\lceil \frac{d_{MAX} \cdot \int_{k_i}^{k_j} \mathcal{D}_{update}(x)dx}{\mathcal{P} := \int_{k_1}^{k_{s_\tau}} \mathcal{D}_{update}(x)dx} \right\rceil \approx d_{MAX} \left\lceil \frac{\int_{k_i}^{k_j} GMM(x, \Psi_\tau)dx}{\mathcal{P}} \right\rceil \tag{6}$$

To approximate $\mathcal{D}_{update}$ using updates $U$, a Gaussian Mixture Model (GMM) (Zhao et al., 2023) is used. The nullifier creates gaps in $D^\tau$ by iterating over successive elements and applying Equation 6. For $j = i + 1$, this simplifies to $\Gamma(k) = GS(k_-, k)$. Nullifer produces NULL values for each $k \in D^\tau$, and the updated index for $k$ is calculated as $GS(k_1, k) \times Index(k)$, with $\Gamma(k)$ vacant spaces before and $\Gamma(k_+)$ after. Figure 4(a) illustrates the procedure, initiated by the Control Unit and relayed to the Retrain module, for constructing a new index model. This occurs after the index ($Y$) is revised by integrating placeholders among the data by utilizing the trained GMM.

## 3.1 TRAINING INCOMING WORKLOAD

The workload updates, $\mathcal{D}_{update}$ distribution, is modeled as a GMM (Section B.3), $\mathcal{D}_{update} \sim$ GMM$(k, \Psi_t)$, where $\Psi_t = \{(\pi_i, \mu_i, \sigma_i)\}_{i=1}^{K}$: weights $\pi_i$, means $\mu_i$, and standard deviations $\sigma_i$. The GMM parameters are generated by a neural network. Training minimizes a loss function balancing model accuracy and simplicity by penalizing unnecessary components.

The initial GMM parameters are established using a greedy approach (see Algorithm 2 in Appendix E). Starting with the data set $D^0$, it groups data into distributions by iteratively forming candidate Gaussians using the two smallest elements and adding points that meet a confidence threshold $\delta$ given as a hyperparameter. Once no more points fit, the cluster is finalized, and the process repeats until all data is assigned. This yields initial parameters $\Psi_0 = \{(\pi_i, \sigma_i)\}_{i=1}^{K}$ and an initial $K$, which may overestimate the true number of components, but provides a strong starting point.

## 3.2 NEURAL UNIT FOR FINE-TUNING.

Figure 2(b) shows the neural architecture $NN_\Psi$ that aims to fine-tune $\psi$ parameters. The model consists of a single fully connected hidden layer with $K$ neurons and leads to an output layer with $3K$ neurons. These layers are set up post-initialization, with the hidden layer being entirely linked to the shared layer $NN_C$. The neural network, initialized with $\Psi_0$ from Algorithm 2, directly predicts the GMM

---

**Algorithm 1** Training Procedure for *Sig2Model* Neural Networks

1: **Input:** Buffer $B^\tau$, Representation $X$, Labels $Y$, Thresholds $\epsilon_\Pi$, $\epsilon_\Psi$, Maximum Iterations $MaxIter$, Neural Network $NN = \{NN_\mathcal{C}, NN_\Pi, NN_\Psi\}$, Sampling Fraction $\lambda$, Replay Buffer Size $\kappa$
2: **Output:** Fine-Tuned $NN$
3: **Initialization:** $iterations \leftarrow 0$, $L_\Pi \leftarrow 2\epsilon_\Pi$, $L_\Psi \leftarrow 2\epsilon_\Psi$
4: $B^\tau_{idx} \leftarrow ReindexRB(B^\tau, RB^{\tau-1})$ ▷ Reindex $RB^{\tau-1}$ using Alg. 3 (Appendix E)
5: **repeat**
6:     $\Pi, \Psi \leftarrow FeedForward(X \cup RB^{\tau-1})$ using updated weights
7:     $L_\Pi \leftarrow \mathcal{L}_\Pi(X \cup B^\tau, Y \cup RB^{\tau-1}.I)$ ▷ Using Equation 5
8:     **if** $L_\Pi \leq \epsilon_\Pi$ and $L_\Psi \leq \epsilon_\Psi$ **then**
9:         **break** ▷ Desired error thresholds achieved
10:     **end if**
11:     **if** $L_\Pi > \epsilon_\Pi$ **then** ▷ $backpropagation^\Pi_{\epsilon_\Pi}$ in Fig 2(b)
12:         Perform backpropagation on $NN_\Pi$
13:         $\Pi, \Psi \leftarrow FeedForward(X \cup RB^{\tau-1})$
14:     **end if**
15:     $L_\Psi \leftarrow \mathcal{L}_\Psi(X \cup B^\tau)$ ▷ Using Equation 7
16:     **if** $L_\Psi > \epsilon_\Psi$ **then** ▷ $backpropagation^\Psi_{\epsilon_\Psi}$ in Fig 2(b)
17:         Perform backpropagation on $NN_\Psi$
18:     **end if**
19:     $iterations \leftarrow iterations + 1$
20: **until** $iterations \geq MaxIter$
21: $RB^\tau \leftarrow UpdateRB(B^\tau_{idx}, RB^{\tau-1}, \lambda, \kappa)$ ▷ Update $RB$ using Alg. 4 (Appendix E)

---

parameters $\Psi = \{(\pi_i, \mu_i, \sigma_i)\}_{i=1}^{K}$. During training, the loss is computed for each batch of input data, gradients are calculated, and network weights are updated via gradient descent. The process continues until the loss converges or GMM parameter changes are negligible. The regularization term ensures components with small weights are suppressed, simplifying the model without explicit pruning.

The GMM parameters are trained using the following loss function:

$$\mathcal{L}_\Psi(X^\tau) = -\sum_{x \in X^\tau} \log\left(\sum_{i=1}^{K} \pi_i N(x \mid \mu_i, \sigma_i)\right) + c\frac{1}{K}\sum_{i=1}^{K} \pi_i^\nu, \tag{7}$$

The first term ensures accurate data modeling $X^\tau$, and the second term penalizes small weights $\pi_i$ to reduce unnecessary components. The regularization strength is controlled by $c > 0$, and $0 < \nu \leq 1$ adjusts the penalty's scaling. This encourages sparsity, which reduces the number of active components.

## 4 SIG2MODEL TRAINING

This section describes the core technical implementation of Sig2Model for the neural network training and model retraining. We first explain the data preparation phase, and then delve into the training processes.

## 4.1 DATA PREPARATION

The data preparation phase combines buffer indices with original data indices to provide the neural network with both embedding representations and updated positional contexts. This integration enables the network to learn patterns based on data features and their positional relationships.

**Buffer and Embedding.** When the buffer, implemented using a B-tree, reaches its capacity $\rho$, nodes are traversed to sort the data and produce $B^\tau$, where $\tau$ represents the buffer's overflow count (stage). Using the current model $M_l'$, the positions of the data points are determined. If a position in $D^\tau$ is already occupied, the data is stored in the buffer. For simplicity, we assume $D^\tau$ is fully occupied, although in practice the training focuses on unaccommodated data.

**Constructing Representation.** Inputs $X^\tau$ are derived from the *normalized numerical features* of current data $D^\tau$ (referred to as $D_{emb}^\tau$), rather than high-dimensional semantic embeddings. Specifically, keys are normalized (e.g., via min-max scaling) to a range suitable for NN processing. First, we obtain $I_{B_{emb}^\tau}$, which represents the buffer indices predicted by $M_l'(\cdot, \Pi_\tau)$. Then, we select the corresponding normalized values from $D_{emb}^\tau$ such that $X^\tau = D_{emb}^\tau[I_{B_{emb}^\tau}]$, ensuring the network trains on the precise positional context of the buffered updates. We have added an example in Section C.2.

**Generating Labels.** Labels $Y^\tau$ account for index changes after updates. For each $X_j^\tau$, the model output index $I_{B_{emb}^\tau}[j]$ is corrected as $Y_j^\tau = I_{B_{emb}^\tau}[j] + j$. Since $B^\tau$ is sorted, the target index is preceded by $j$ new update indices, ensuring accurate model training.

## 4.2 TRAINING PROCESS

The training process operates in two phases: *(1) updating neural networks for incoming updates and SigmaSigmoid modeling, (2) evolving the model from $M_l$ to $M_{l+1}$*. These phases improve system representation through the integration of $\mathcal{N}$ sigmoid functions within the neural network while maintaining prior data buffers via $\sum_{i=1}^{\mathcal{N}} \sigma(., A_i, \omega_i, \phi_i)$. The system triggers updates to the original index model when performance metrics indicate degradation.

**Neural Network Training Process.** This section describes the neural network training procedure (Figure 2(b)). Two cost functions, $\mathcal{L}_\Pi$ and $\mathcal{L}_\Psi$, are computed based on the outputs of $NN_\Pi$ and $NN_\Psi$. As these operate in different dimensions of spaces, a specialized training strategy (Algorithm 1) is used. Each model minimizes its cost independently, iterating until both $L_\Pi \leq \epsilon_\Pi$ and $L_\Psi \leq \epsilon_\Psi$ are met or the maximum iterations are reached.

The training process starts with a *feed forward* step using data from the current buffer $B^\tau$ (Section 4.1). Errors are calculated and backpropagation is performed only if $L_\Pi > \epsilon_\Pi$ or $L_\Psi > \epsilon_\Psi$. Priority is given to $NN_\Pi$ if its error exceeds the threshold. The weights are updated, and the cycle continues with $NN_\Psi$, using a batch size equal to the buffer size $\rho$. Training stops when both errors are below their respective thresholds or when the iteration limit is reached. Both cost functions have closed-form derivatives for efficient optimization.

**Learned Index Model Retraining.** This section explains the *Retrain Module* for the underlying LI model, shown in Figure 4(b), focusing on retraining signals, data preparation, and system re-initialization after retraining for new updates.

*Retrain Signals.* Signals for retraining come from *NN Training* and the *Inference Module* (Figure 2(b,c)). During training, signals arise when backpropagation reaches its limit ($iterations \geq MaxIter$ in Algorithm 1). During inference, signals occur when out-of-range searches show that $M_l'$ struggles to maintain accuracy, typically when the system reaches maximum capacity (Section G.2), and the target not be found in the $\epsilon$-bounded range.

*Data Preprocessing.* We sort the data for training a new LI model. The output of $NN_\Psi$ generates a GMM optimized for the update workload distribution ($\mathcal{D}_{update}(k) \sim GMM(k, \Psi_\tau)$). After the buffer reaches capacity, $B^\tau$ is merged with $D^\tau$ to create $D^{\tau+1}$. Using $\mathcal{D}_{update}$, Equation 6 introduces gaps between data points. Training data for $M_{l+1}$ are formed by pairing the entries in $D^{\tau+1}$ with their new indices. This ensures that the model smoothly handles anticipated gaps without need for modifications.

*Neural Networks Re-Initialize.* Upon receiving update signals (Figure 4), the Control Unit initiates training for the new model $M_{l+1}$ while executing a systematic re-initialization protocol. The SigmaSigmoid parameters undergo complete reset to preserve the update distribution $\mathcal{D}_{update}$, with

Table 1: **QPS comparisons with state-of-the-art methods. The numbers are in $10^6$.QPS (MQPS).**

| Workload | Dataset | S2M | | S2M−Ψ | | S2M−B | | B+Tree | Alex | LIPP | DILI | Average | | Max | |
|---|---|---|---|---|---|---|---|---|---|---|---|---|---|---|---|
| | | S | M7 | S | M7 | S | M7 | | | | | S | M | S | M |
| Read-Only | Wiki | **5.4** | 5.4 | NA | NA | NA | NA | 2.0 | 4.1 | 5.2 | **5.4** | 50.5% | 48.8% | 2.68x | 2.68x |
| | Logn | **6.3** | 6.2 | NA | NA | NA | NA | 1.9 | 6.2 | 6.0 | **6.3** | 56.1% | 55.4% | 3.18x | 3.18x |
| | FB | **4.5** | 4.5 | NA | NA | NA | NA | 2.0 | 2.3 | 4.3 | **4.5** | 55.5% | 51.6% | 2.23x | 2.23x |
| Read-Heavy | Wiki | **4.3** | 5.4 | 4.1 | 5.1 | 3.9 | 4.9 | 1.5 | 2.2 | 3.9 | 4.1 | 69.3% | 75.8% | 2.86x | 3.41x |
| | Logn | **4.1** | 5.1 | 3.9 | 4.9 | 3.8 | 4.7 | 1.5 | 3.2 | 2.9 | 4.0 | 61.2% | 70.1% | 2.71x | 3.38x |
| | FB | **2.6** | 3.3 | 2.5 | 3.1 | 2.4 | 3.0 | 1.3 | 1.1 | 2.3 | 2.5 | 62.8% | 72.4% | 2.28x | 2.91x |
| Write-Heavy | Wiki | **3.6** | 4.5 | 3.3 | 4.2 | 3.1 | 4.0 | 1.3 | 1.9 | 2.8 | 3.1 | 73.8% | 80.2% | 2.76x | 3.45x |
| | Logn | **3.9** | 4.8 | 3.7 | 4.6 | 3.4 | 4.3 | 1.3 | 3.2 | 2.9 | 3.6 | 76.2% | 83.5% | 3.00x | 3.70x |
| | FB | **3.9** | 4.7 | 3.6 | 4.4 | 3.3 | 4.1 | 1.3 | OOM | 2.1 | 3.5 | 82.2% | 88.4% | 3.00x | 3.61x |
| Write-Only | Wiki | **2.9** | 2.9 | 2.7 | 2.8 | 2.4 | 2.4 | 1.3 | 1.4 | 2.2 | 2.4 | 76.8% | 76.8% | 2.23x | 2.23x |
| | Logn | **3.1** | 3.2 | 3.0 | 3.0 | 2.6 | 2.6 | 1.2 | 2.8 | 2.2 | 2.5 | 58.4% | 62.8% | 2.58x | 2.66x |
| | FB | **2.5** | 2.5 | 2.2 | 2.2 | 2.0 | 2.0 | 1.2 | OOM | 1.7 | 2.1 | 68.1% | 68.1% | 2.04x | 2.04x |

distinct handling procedures for each neural network component: $NN_\Psi$ remains unchanged, continuously adapting to incoming data streams. For $NN_\Pi$, the system neutralizes sigmoids from the previous model $M_l$ through four coordinated operations: (1) complete purging of the replay buffer, (2) zero-initialization of weights connecting to the output neurons $\{A_j\}_{j=1}^{\mathcal{N}}$, (3) recalibration of sigmoid parameters $\{\phi_j\}_{j=1}^{\mathcal{N}}$ using either $\mathcal{D}_{\text{update}}$ or uniform random sampling from the key space, and (4) uniform weight adjustment setting $\omega_j = 1$ for all $j \in [1, \mathcal{N}]$. Uniformly modifying weights to set $\omega_j = 1|_{j=1}^{\mathcal{N}}$. The neural network $NN_{\mathcal{C}}$ remains unchanged, preserving the feedforward paths from $NN_{\mathcal{C}}$ to $NN_\Psi$ to maintain the GMM while resetting SigmaSigmoid for future updates.

## 5 EXPERIMENTAL EVALUATION

We conduct a comprehensive evaluation of Sig2Model against state-of-the-art learned indexes (DILI, LIPP, and Alex). Sig2Model shows significant improvements across three key metrics: Up to $3\times$ higher QPS, $20\times$ lower training cost, and $1000\times$ reduced memory usage compared to baseline methods. These improvements are particularly significant in update-intensive scenarios that highlight Sig2Model's architectural advantages. Experiments were conducted using a 40GB vRAM GPU for background training. Complete experimental configurations (hardware specs and baselines) and additional results are available in Appendices H and I respectively.

**QPS Comparison.** Table 1 presents a detailed QPS comparison between Sig2Model (S2M), three baseline methods, and two ablated variants: S2M−Ψ (without placeholder training) and S2M−B (without buffer component). We evaluate both single-threaded and multi-threaded configurations of Sig2Model variants. We provide the details on multi-threading in Appendix I.3. Sig2Model shows superior QPS scaling as update rates increase, achieving an 82% average improvement (calculated as the geometric mean of speedups across all datasets and workloads) over baseline methods. This advantage is most distinct in the FB dataset (Write-Heavy), where baselines like LIPP degrade significantly or OOM.

In read-only workloads, Sig2Model matches the performance of its underlying RadixSpline implementation while outperforming B+Tree by 15-20% and achieving comparable results to DILI. The ablated variants S2M−Ψ and S2M−B are excluded from these tests, as they specifically optimize update handling rather than read performance. For read-heavy workloads with 10% updates, Sig2Model achieves $2.7\times$ higher QPS on the Logn dataset, with multi-threading providing $3.4\times$ speedup. As update rates increase, Sig2Model maintains a consistent 60% average QPS advantage over competing methods. Write-heavy workload tests reveal particularly strong performance, with Sig2Model processing $4.7$ MQPS on the Facebook dataset, significantly outperforming B+Tree (1.3M), LIPP (2.1M), and DILI (3.5M). Alex also fails in several experiments due to its known memory constraints (Yang et al.). In write-only scenarios, Sig2Model achieves 67.7-74.5% higher QPS than baselines through its optimized update handling mechanisms.

**Various Request Distribution.** Figure 5(a) demonstrates Sig2Model's consistent performance across

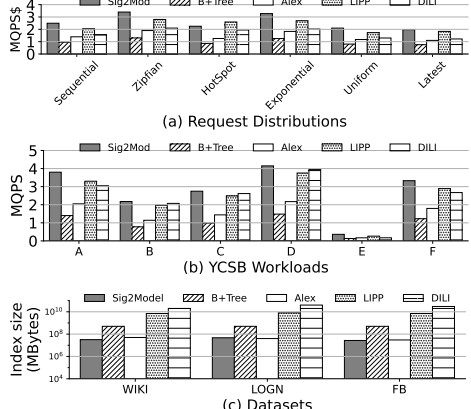

(a) Request Distributions

(b) YCSB Workloads

(c) Datasets

Figure 5: **Evaluation results. (a) QPS on request distributions. (b) YCSB workloads. (c) Index memory sizes.**

Table 2: **Components ablation study**

| Components | None | **B** | Ψ | Π | **B**+Ψ | Π+Ψ | **B**+Π | Full |
|---|---|---|---|---|---|---|---|---|
| MQPS | 0.03 | 1.4 | 2.0 | 2.1 | 2.3 | 2.9 | 3.0 | 3.6 |

six different request distributions for write-heavy workloads using the Wiki dataset (Dataset, 2019b). The system maintains QPS improvements of $2.61\times$ over B+Tree, $1.78\times$ over Alex, $1.15\times$ over LIPP, and $1.54\times$ over DILI, showing a robust adaptation to varying access patterns. Note that while DILI outperforms LIPP on skewed (Zipfian) workloads (Table 1) due to caching, LIPP often surpasses DILI on Uniform distributions (Fig. 5a) where tree-structure overheads dominate.

**YCSB Benchmark Results.** Figure 5(b) shows Sig2Model achieves consistently superior QPS across all six YCSB workloads (Cooper et al., 2010). For read-intensive workloads (YCSB B, C, and D), the system delivers 2.1-4.1 MQPS, outperforming baseline methods by $1.0$-$2.8\times$. In balanced workloads (YCSB A and F), Sig2Model maintains strong performance at 3.3-3.8 MQPS while sustaining a 55.1% average QPS advantage. The system particularly excels in scan-heavy operations (YCSB E), where its efficient range query processing yields 55% higher QPS than DILI and significantly outperforms other baselines that suffer from substantial re-traversal overhead.

**Index Memory Size.** Figure 5(c) shows the memory usage of the Sig2Model and the baselines, including memory for fine-tuning neural networks (weights, replay buffers) in the Sig2Model. Sig2Model uses up to $1000\times$ less memory than DILI; DILI's memory consumption explodes due to the materialization of new physical tree nodes for every split, whereas Sig2Model maintains a compact, fixed-size architecture due to its efficient placeholder placement. In contrast, B+Tree consumes significantly more memory due to the storage of all keys in leaves and the inherent tree structure overhead (pointers and fill-factor). Similarly, LIPP consumes high memory due to empty slots created during conflicts. Alex's in-place placeholder strategy is less efficient as it does not consider the incoming workload distribution, leading to slightly higher memory usage than Sig2Model.

**Index Memory Size.** Figure 5(c) shows the memory usage of the Sig2Model and the baselines, including memory for fine-tuning neural networks in the Sig2Model. Sig2Model uses up to $1000\times$ less memory than DILI, due to its efficient placeholder placement and minimized buffer additions. In contrast, B+Tree consumes significantly more memory due to the storage of all keys in leaves and the inherent tree structure overhead (pointers and fill-factor). Similarly, DILI and LIPP consume high memory due to new leaf nodes and empty slots created during conflicts. Alex's in-place placeholder strategy is less efficient as it does not consider the incoming workload distribution, leading to slightly higher memory usage than Sig2Model.

**Core Components Ablation Study.** We analyze the individual contributions of the three core components of Sig2Model on the Wiki dataset under write-heavy workloads: (1) buffer management (**B**), (2) index approximation network using SigmaSigmoid boosting (Π), and (3) update workload training using GMM (Ψ). Table 2 presents QPS measurements for all combinations of components. The baseline RadixSpline with full retraining (*None*) achieves only 0.03 MQPS. Individual components show varying effectiveness: Ψ (2.0 MQPS, comparable to Alex), Π (2.1 MQPS), and **B** (1.4 MQPS). The combinations of two components show synergistic effects, with **B**+Π achieving 3.0 MQPS and Π+Ψ reaching 2.9 MQPS (surpassing LIPP). The complete Sig2Model configuration (**B**+Π+Ψ) delivers optimal performance at 3.6 MQPS, consistent with our full system results in Table 1.

**Training Loss Curves.** Figure 6(a) shows the loss curves for $NN_\Pi$ and $NN_\Psi$ during ten rounds of fine-tuning on the Wiki dataset using the read-heavy workload. Both networks train smoothly and converge to $\epsilon_\Pi$ and $\epsilon_\Psi$. Initially, $NN_\Pi$ requires about 50 epochs to converge, but this decreases to 29 epochs in later updates due to memory retained from $NN_C$. Similarly, $NN_\Psi$ requires fewer epochs in subsequent updates, as the update distributions remain consistent,

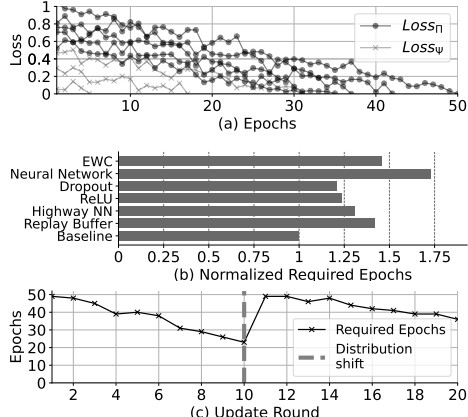

Figure 6: **(a) Loss curves for $NN_\Pi$ and $NN_\Psi$. (b) Ablation study on $NN_c$. (c) Impact of distribution shift on required epochs.**

and the GMM parameters ($\Psi$) do not require significant changes. After four updates with stable workload, the initial loss for $NN_\Psi$ falls below $\epsilon_\Psi$, eliminating the need for further iterations.

$NN_C$ **Ablation Study.** Figure 6(b) analyze the contributions of $NN_C$ components to training efficiency by measuring the increase in epochs required to reach $\epsilon_\Pi$ and $\epsilon_\Psi$ when components are removed. Removing the reply buffer, highway NN, ReLU, Dropout, and EWC increase the epochs by $1.42\times$, $1.31\times$, $1.24\times$, $1.21\times$, and $1.46\times$, respectively. Simplifying fully connected layers from 3 to 1 results in a $1.73\times$ increase in epochs.

**Retrain Cost Analysis.** Figure 7 shows retraining costs for Sig2Model and baselines under a write-only workload on the Wiki dataset. Sig2Model achieves up to $2.20\times$ reduction in the total number of retrainings and a $20.58\times$ decrease in the total duration of retraining compared to the baselines.

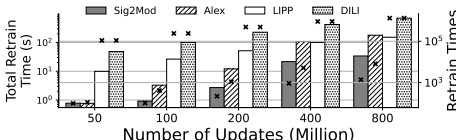

Figure 7: **Retrain cost**

**Limitations.** *(1) Distribution Shift.* Since the parameters in SigmaSigmoids and GMM are trained on an initial data distribution, any shift in the distribution requires additional training time for the neural network to adapt. For example, as shown in Figure 6(c), the number of training epochs spikes during the 11th round when the data distribution changes from Zipfian to Exponential. *(2) Fixed SigmaSigmoid Capacity ($\mathcal{N}$).* In our experiments, we use a constant value for $\mathcal{N}$ based on hyperparameter sensitivity test (Adkins et al., 2024)(See Appendix I.4). However, this value can be dynamically adjusted based on the observed data distribution and workload to improve further improve the performance.

## 6 CONCLUSION

Sig2Model presents a mathematically rigorous framework for efficient learned index updates, directly addressing the critical retraining bottleneck through innovative model adaptation techniques. While index updates remain inherently non-local operations, our approach guarantees bounded sub-optimality. By employing a boosting methodology, Sig2Model demonstrates that an ensemble of weak approximators can progressively converge toward an optimal update policy—eliminating the need for expensive full retraining cycles. While demonstrated on RadixSpline, the SigmaSigmoid boosting framework is model-agnostic and can be applied to other learned indexes (e.g., PGM-index) to enable updatability. This fundamental advancement not only maintains index effectiveness during updates but also opens new research directions for sustainable learned index architectures. Our work establishes a foundation for future systems that can adapt dynamically to workload changes while preserving theoretical guarantees.

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

## APPENDIX CONTENTS

# A Table of Notations

Table 3: Notations

| | |
|---|---|
| $D^\tau$ | Sorted keys after $\tau^{\text{th}}$ update with size $s_\tau$ |
| $D^\tau$ | Sorted keys after $\tau^{\text{th}}$ update with size $s_\tau$ |
| $D_{emb}^\tau$ | Embedded keys of $D^\tau$ with vectors of size $n$ |
| $B^\tau$ | The single buffer for entire index structure at stage $\tau$ |
| $\rho$ | The size of the buffer |
| $RB^\tau$ | The neural network replay buffer at stage $\tau$ |
| $\kappa$ | The size of $RB^\tau$ |
| $X_i^\tau$ | $i^{\text{th}}$ input to the neural network at stage $\tau$ |
| $Y_i^\tau$ | Corresponding label of $X_i^\tau$ |
| $A$ | Amplitude of Sigmoid function |
| $\omega$ | Slope of Sigmoid function |
| $\phi$ | Center of Sigmoid function |
| $\sigma(x, A, \omega, \phi)$ | Parametrized Sigmoid function |
| $\mathcal{N}$ | Maximum number of Sigmoids |
| $\mathcal{D}_{update}$ | The distribution of incoming updates |
| $U$ | Set of new updates drawn from $\mathcal{D}_{update}$ |
| $M_l(.)$ | The LI model after $l^{\text{th}}$ retrain with maximum error $E$ |
| $M_l'(., \Pi_\tau)$ | The adjusted model of $M_l(.)$ with parameter $\Pi_\tau$ at stage $\tau$ |
| $E_l$ | The maximum estimation error of $M_l(.)$ |
| $\lambda$ | Sampling fraction |
| $\Pi$ | Set of sigma sigmoid parameters |
| $\Psi$ | Set of GMM parameters |
| $K$ | Number of GMM's kernel |
| $\mathbb{M}$ | The hypothesis space of Sigma-Sigmoid based models, all configurations of $\mathcal{N}$ sigmoids, with parameters defined by $\Pi$. |
| $\Gamma(k)$ | Determine size gap before given $k$ |
| $G$ | Maximum gap between continuous keys |
| $d$ | Minimum possible distance between keys |
| $\alpha$ | Model prediction bias factor |
| $\beta$ | Interference factor |
| $\delta$ | Confidence level of initial clustering |
| $E_\Pi$ | Prediction confusion parameter |
| $\epsilon_\Pi / \epsilon_\Psi$ | Error threshold for Sigma-Sigmoid/Incoming updates model |
| $s$ | Size of data |

# B Preliminaries

## B.1 Learned Index (LI)

The learned indexes (Ding et al., 2020; Chatterjee et al., 2024; Li et al., 2020; Tang & et al., 2020; Kipf et al., 2020; Kim et al., 2024; Lan et al., 2024) aim to improve the efficiency of data retrieval in database systems by using machine learning models that map keys to their locations. The traditional learned index employs ensemble learning and hierarchical model organization. Starting from the root node and progressing downward, the model predicts the subsequent layers to use for a query key $k$ based on $F(k) \times s$, where $s$ is the number of keys, and $F$ is the cumulative distribution function (CDF) that estimates the probability $p(x \leq k)$. Given the overhead of training and inference in complex models, most learned indexes utilize piecewise linear models to fit the CDF. Querying involves predicting the key's position using $pos = a \times k + b$ with a maximum error $e$, where $a$ and $b$ are learned parameters, and $e$ is for the final search to locate the target key.

**Updatable Learned Index.** Learned indexes require a fixed record distribution, making updates difficult (Section 1). Solutions for up-datable learned indexes include: *(i) delta buffer, (ii) in place, (iii) hybrid structures* Sun et al. (2023); Ge & et al. (2023). *Delta buffer* methods (e.g., LIPP) use buffers to postpone updates, but merging occurs when buffers overflow. *In-place* approaches (e.g., Alex) reserve placeholders for updates but may cause inefficient searches when offsets fill up. *Hybrid*

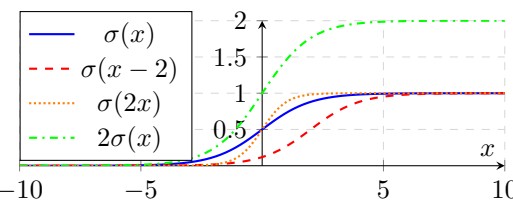

Figure 8: Sigmoid function $\sigma(x)$ transformations.

methods balance efficiency and speed by combining buffers and placeholders. DILI, a hybrid solution, uses a tree structure for level-wise lookups, but updates increase the tree height over time.

## B.2 MODEL ADJUSTMENT WITH SIGMOIDS

As discussed in Section 1, sigmoid functions enable smooth model adjustments by treating small updates as gradual changes, reducing the frequency of retraining. The sigmoid function $\sigma(x) = \frac{1}{1+e^{-x}}$ creates a "S"-shaped curve, commonly used in machine learning.

When combined with another function, for example, $f(x) + \sigma(x - \phi)$, it introduces a smooth step-like transition near $\phi$. This property makes sigmoids ideal for approximating stepwise behaviors.

The generalized sigmoid, $\sigma(x, A, \omega, \phi) = \frac{A}{1+e^{-\omega(x-\phi)}}$, adds flexibility to control amplitude, slope, and center, enabling broader behavior modeling in learned indexes.

## B.3 GAUSSIAN MIXTURE MODEL FOR DISTRIBUTION MODELING

A *Gaussian Mixture Model* (GMM) assumes data are generated from a mixture of Gaussian distributions, each defined by a mean and variance. GMMs are flexible and well-suited for modeling complex, multimodal key distributions in real-world workloads. The GMM is mathematically expressed as

$$\text{GMM}(x, \Psi) = p(x) = \sum_{i=1}^{K} \pi_i N(x \mid \mu_i, \Sigma_i) \tag{8}$$

where $K$ is the number of Gaussian components (kernels) and $\Psi = \{(\pi_i, \mu_i, \sigma_i)\}_{i=1}^{K}$ represents the GMM parameters, $\pi_i$ is the weight of the $i$-th component, and $N(x \mid \mu_i, \Sigma_i)$ is the Gaussian distribution with mean $\mu_i$ and covariance $\Sigma_i$ ($\sum_{i=1}^{K} \pi_i = 1$).

In learned indexes, GMMs predict update distributions. Each Gaussian component represents a key cluster, enabling the precise placement of placeholders for future updates.

## B.4 JUSTIFICATION FOR SIGMOIDAL BASIS FUNCTION SELECTION

We explicitly justify the selection of sigmoid functions as the basis for modeling update residuals ($\Pi$) over other function classes (e.g., Piecewise Linear (PWL), Polynomials, or Radial Basis Functions) based on three fundamental properties:

**1. Inductive Bias and Step-wise Nature.** An insertion update at key $u$ conceptually introduces a *Heaviside step function* error distribution: it adds a displacement of 0 for keys $k < u$ and $+1$ for keys $k > u$. Polynomials (due to oscillation/Runge's phenomenon) and Radial Basis Functions (due to their local decay) fail to capture this semi-global "step" behavior efficiently. The sigmoid function is the canonical smooth approximation of the Heaviside step. It provides the correct inductive bias for the problem, capturing the global shift of the tail ($k > u$) with a single term, whereas local methods (like PWL or splines) would require updating parameters for every subsequent segment to maintain the $\epsilon$-bound.

**2. Differentiability for Neural Optimization.** A critical requirement for our Neural Joint Optimization (Section 2) is a loss landscape suitable for gradient descent. While a raw step function is non-differentiable at the jump, and PWL functions have non-differentiable "kinks" at boundaries, the sigmoid is smooth ($C^\infty$) and differentiable everywhere. This smoothness ensures stable, non-zero gradients $\nabla_\Pi \mathcal{L}$ during backpropagation, allowing the neural tuner ($NN_\Pi$) to precisely regress the location ($\phi$) and steepness ($\omega$) of updates.

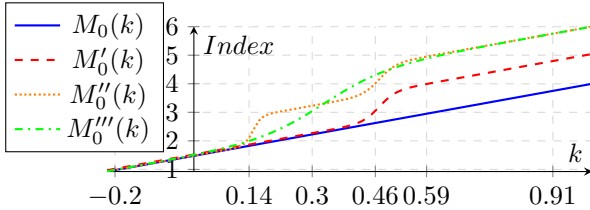

Figure 9: Adjusting the LI model upon receiving updates by employing the sigmoid as a step function.

**3. Efficiency in Boosting Ensembles.** In the context of boosting, we utilize sigmoids as "weak learners." Because a single sigmoid naturally models the cumulative effect of an insertion, the ensemble sum $O(\mathcal{N})$ converges significantly faster than linear approximations, which must approximate a step using multiple segments (high slope linear pieces). As shown in our sensitivity analysis, a small ensemble ($\mathcal{N} = 20$) is sufficient to model complex update patterns, keeping inference overhead negligible compared to the $O(\log N)$ tree-search costs.

## C    EXAMPLES AND WALKTHROUGHS

### C.1    MOTIVATIONAL EXAMPLE

Consider the LI model $M_0(k) = 2.5k + 1.5$ with a prediction error of $E = 0.25$ in the keys domain $D^0 = \{-0.2, 0.3, 0.59, 0.91\}$. The predicted indices for the elements in $D^0$ yield $I_{D^0} = M_0(D^0) = \{1, 2.25, 2.975, 3.775\}$. Consequently, for each $I \in I_{D^0}$, the range of search positions is given as $I \pm E$. For example, for the key $k = 0.59$ where the prediction is $M_0(0.59) = 2.975$, the search range is $2.975 \pm 0.25$, resulting in the interval $[2.725, 3.225]$. Considering that positions are integers, we only search for positions $3$ and $4$.

Consider an incoming update $u_1$ with a key value of $u_1 = 0.46$. This update alters the domain, $D^1 = \{-0.2, 0.3, 0.46, 0.59, 0.91\}$, while the index set $I_{D^1} = \{1, 2.25, 2.65, 2.975, 3.775\}$. The update does not affect the indices for $-0.2$ and $0.3$, so the model $M_0$ remains applicable. However, for the elements $2.975$ and $3.775$, the indices are outdated. In essence, the predictions for elements $\leq u_1$ remain unchanged, while elements in the domain that are $\geq u_1$ are impacted. Furthermore, it is evident that elements $> u_1$ experience an exact effect of $1$, implying that incrementing their previous prediction by one aligns their indices with the new index. This insight suggests that by implementing a step-like adjustment to $M_0$, we can avoid training a new model $M_1$ across $D^1$. By adding $\frac{1}{1+e^{-48.3(k-0.47)}}$ to $M_0$, a new model is generated that produces adjusted outputs $M_0'(k) = M_0(k) + \left(\frac{1}{1+e^{-48.3(k-0.47)}}\right)$. Then, $I_{M_0'(D^1)} = \{1, 2.25, 3.03, 3.971, 4.775\}$ which using $E$ gives us the correct index for all the keys in $D^1$. Given the second update $u_2 = 0.14$, $D^2 = \{-0.2, 0.14, 0.3, 0.46, 0.59, 0.91\}$, you might choose to apply another step function, a second sigmoid, expressed as $M_0''(k) = M_0'(k) + \left(\frac{1}{1+e^{-98.7(k-0.15)}}\right)$, or you can fully utilize the capacity of the initial sigmoid applied in $M_0'$, formulated as $M_0'''(k) = M_0(k) + \left(\frac{2}{1+e^{-12.6(x-0.33)}}\right)$. In line with the *Oakum razor principle*, we choose $M_0'''$ instead of $M_0''$ because it is simpler and needs less memory. However, the minimal memory usage isn't always feasible, as it relies on the interval between (i.e., distribution) updates. Consequently, the suggested model should account for these intervals when determining the count of step functions (i.e., sigmoids). These approximations are shown in Figure 9. This example demonstrates that by employing an appropriate step function, we can modify the LI model without the need to retrain completely on new data. In subsequent sections, we expand this concept and establish a formal learning framework to train the step function.

### C.2    DATA PREPARATION WALKTHROUGH

To clarify the data preparation process described in Section 4.1, we provide a concrete numerical example. Consider a sorted dataset $D^\tau = [10, 20, 30, 40]$ and a global key domain of $[0, 100]$.

**Normalization ($D_{emb}^\tau$):** The keys are normalized (e.g., divided by 100) to create numerical features suitable for the NN: $D_{emb}^\tau = [0.1, 0.2, 0.3, 0.4]$.

**Buffer Processing:** Assume the buffer contains a new update $u = 25$. The current model $M_l'$ predicts the position of $u$ within the existing data. If $M_l'$ maps $u$ to index 2 (corresponding to key 20), then $I_{B_{emb}^\tau} = 2$.

**Input Construction** ($X^\tau$)**:** The network input is derived from the normalized feature of the predicted position: $X^\tau = D_{emb}^\tau[2] = 0.2$.

**Label Generation** ($Y^\tau$)**:** The label represents the corrected position. Since $u = 25$ should be inserted after 20, the target index is adjusted to reflect the sorted order.

This process ensures the neural network learns to correct the position of updates relative to the normalized distribution of existing keys.

## D   CANONICAL NEURAL NETWORK ($NN_C$)

This section describes the shared weights of $NN_C$, a core component of the neural networks linked to system parameters and update workload distribution. $NN_C$ processes embedded data to build knowledge memory and representations for $NN_\Pi$ and $NN_\Psi$. The next section details the architecture of $NN_C$, followed by an explanation of data preparation for its input and the training process for these networks during initialization and mid-development.

The multi-layer neural network $NN_C$ processes sequential data batches for continuous learning. It employs highway networks, ReLU activations, and dropout layers to extract patterns and prevent overfitting. To address catastrophic forgetting Kemker et al. (2018), two strategies are used: *Replay Buffer* Di-Castro et al. (2022) and *Elastic Weight Consolidation (EWC)* Kirkpatrick et al. (2017). The network outputs feed into two subnets, $NN_\Psi$ and $NN_\Pi$, each handling specific tasks. Key components of $NN_C$ include:

**Replay Buffer** ($RB$)**.** The replay buffer stores observed data ($D^0$, $B^1$, $B^2$, ..., $B^{\tau-1}$) as 4-ary tuples: key, representation, global index, and age. Algorithm 3 updates global indexes for $RB^{\tau-1}$ upon new data $B^\tau$, ensuring $k_{MIN}$ and $k_{MAX}$ are preserved the updated . Algorithm 4 refreshes the replay buffer after neural networks by adding new data or replacing the oldest entries probabilistically when capacity $\kappa$ is reached.

**Highway Neural Network (2 layers).** Input data (new and replayed) is processed through a 2-layer highway network to mitigate vanishing gradients, similar to residual connections Srivastava et al. (2015). This approach preserves essential features while learning new ones Wang et al. (2024).

**Nonlinear Activation (ReLU).** ReLU activation introduces non-linearity essential for capturing complex relationships and avoids vanishing gradients during backpropagation Li & Yuan (2017); Ribeiro et al. (2020).

**Dropout Layer.** Dropout randomly disables neurons during training to prevent overfitting and improve generalization Srivastava et al. (2014); Baldi & Sadowski (2013).

**Neural Network (3 layers).** A 3-layer network improves the understanding of complex data relationships, helping generalization between tasks.

**EWC Regularization.** Elastic Weight Consolidation (EWC) Liu et al. (2020) protects critical parameters during training, mitigating catastrophic forgetting.

**Fully Connected to Subnets.** The final layer connects to subnets $NN_\Psi$ and $NN_\Pi$, each handling specific tasks, improving performance in multi-task scenarios.

# E   ALGORITHMS

---

**Algorithm 2** Greedy Initialization of GMM Parameters

---

1: **Input:** Data $D^0$ with Size $s_0$, Confidence Level $\delta$
2: **Output:** GMM Parameters $\Psi_0$, Number of Kernels $K$
3: $\Psi_0 \leftarrow \emptyset$, $K \leftarrow 0$
4: **while** $|D^0| \geq 2$ **do**
5:      Select the two smallest elements: $k_1, k_2 \in D^0$
6:      $T \leftarrow \{k_1, k_2\}$, $D^0 \leftarrow D^0 \setminus \{k_1, k_2\}$
7:      Construct a Gaussian $N_T = N(\mathrm{avg}(T), \mathrm{samplevar}(T))$
8:      **for** each element $k \in D^0$ **do**
9:          **if** $P(k \sim N_T) > \delta$ **then**
10:             $T \leftarrow T \cup \{k\}$
11:             Update $N_T \leftarrow N(\mathrm{avg}(T), \mathrm{samplevar}(T))$
12:             $D^0 \leftarrow D^0 \setminus \{k\}$
13:          **else**
14:             $\Psi_0 \leftarrow \Psi_0 \cup \left( \frac{|T|}{s_0}, \mathrm{avg}(T), \mathrm{samplevar}(T) \right)$
15:             $K \leftarrow K + 1$
16:          **end if**
17:      **end for**
18: **end while**
19: **if** $|D^0| \neq 0$ **then**
20:      Add remaining elements to $T$
21:      Update the last normal distribution and its coefficient
22: **end if**

---

**Algorithm 3** $ReindexRB()$ ReIndex Replay Buffer Indexes

---

1: **Input:** Updates Buffer $B^\tau$, Replay Buffer $RB^{\tau-1} = \{(k_1, r_1, I_1, a_1), \ldots, (k_\kappa, r_\kappa, I_\kappa, a_\kappa)\}$
2: **Output:** Reindexed Replay Buffer $RB^{\tau-1}$, Indexed Buffer $B^\tau_{idx}$
3: $cntr \leftarrow 1$, $B^\tau_{idx} \leftarrow \emptyset$
4: **for** $i = 1$ to $\kappa$ **do**
5:      **while** $cntr \leq |B^\tau|$ and $B^\tau[cntr].key < k_i$ **do**
6:          $Add((B^\tau[cntr].key, B^\tau[cntr]_{emb}, I_i + cntr, 0))$ to $B^\tau_{idx}$
7:          $cntr \leftarrow cntr + 1$
8:      **end while**
9:      $I_i \leftarrow I_i + cntr$
10:      $a_i \leftarrow a_i + 1$             ▷ Increment age excluding $k_{MIN}$ and $k_{MAX}$
11: **end for**
12: **while** $cntr \leq \rho$ **do**
13:      $Add((B^\tau[cntr].key, B^\tau[cntr]_{emb}, I_m + cntr, 0))$ to $B^\tau_{idx}$
14:      $cntr \leftarrow cntr + 1$
15: **end while**

---

**Algorithm 4** $UpdateRB()$ Update Replay Buffer from $\tau - 1$ to $\tau$

---

1: **Input:** Indexed Buffer $B^\tau_{idx}$, Replay Buffer $RB^{\tau-1}$, Sampling Fraction $\lambda$, Replay Buffer Size $\kappa$
2: **Output:** New Replay Buffer $RB^\tau$
3: **for** each entry $e$ in $B^\tau_{idx}$ with probability $\lambda$ **do**
4:      **if** $|RB^{\tau-1}| < \kappa$ **then**
5:          Add $e$ to $RB^{\tau-1}$
6:      **else**
7:          Replace oldest element in $RB^{\tau-1}$ with $e$
8:      **end if**
9: **end for**
10: $RB^\tau \leftarrow SortByKey(RB^{\tau-1})$

---

# F    PROOF OF THEOREM

We provide the proof for Theorem 1 in this section.

**Proof 1** *Step 1 (Discrete Spacing).* *By hypothesis, for any two consecutive keys $k_i < k_{i+1}$, the gap satisfies $1 \leq \left| M'(k_{i+1}, \Pi) - M'(k_i, \Pi) \right| \leq 2$. Hence, if we list keys $k$ in ascending order, each key $k$ can have only a small number of "neighboring keys" $k_{\pm}$ for which $|M'(k, \Pi) - M'(k_{\pm}, \Pi)| < 2$. In particular, for $E_{\Pi}$ chosen in the range $[1, 2)$, only the same key $k$ or its one or two immediate neighbors in the sorted order of keys can satisfy $|M'(k, \Pi) - M'(k_{\pm}, \Pi)| < E_{\Pi}$.*

*Step 2 (Choose $E_{\Pi} \geq 1$).* *We pick $E_{\Pi} \geq 1$. Because each key $k$ has at most $O(1)$ neighboring keys whose model outputs lie within $E_{\Pi}$, the event $\left| M'(k, \Pi) - M'(u, \Pi) \right| < E_{\Pi}$ can only occur if $u$ is either $k$ itself or one of those few neighbors. Let $Neighbor(k, E_{\Pi})$ denote this (small) set of possible neighbors of $k$. Then $\left| M'(k, \Pi) - M'(u, \Pi) \right| < E_{\Pi} \implies u \in \{k\} \cup Neighbor(k, E_{\Pi})$.*

*Step 3 (Bound the Probability).* *Since the set $\{k\} \cup Neighbor(k, E_{\Pi})$ is small and fixed for each $k$, the probability that a random $u \sim \mathcal{D}_{update}$ lands in that set is bounded above by some function of its measure or cardinality. Specifically,*

$$\mathbb{P}\Big[ \big| M'(k, \Pi) - M'(u, \Pi) \big| < E_{\Pi} \Big] \leq \mathbb{P}\Big[ u \in \{k\} \cup Neighbor(k, E_{\Pi}) \Big].$$

*Under assumption bounded $\mathcal{D}_{update}$, there exist a finite $\beta$ and we can make this probability $\leq \beta$ (e.g., $\beta$ can be chosen based on density).*

*Thus, there is an $E_{\Pi} \geq 1$ such that $\mathbb{P}\big[ \big| M'(k, \Pi) - M'(u, \Pi) \big| < E_{\Pi} \big] \leq \beta$, completing the proof.*

# G    THEORETICAL ANALYSIS

This section addresses two theoretical aspects of Sig2Model: (1) how individual updates affect $\omega$ in the sigmoid approximation and (2) the neural network's feasibility in achieving optimal sigmoid-based approximation. We define $\epsilon$ as the maximum error within $M$'s domain caused by the sigmoids' gradual transition from $0$ to their maximum $A$. Lastly, we analyze the time complexity of Sig2Model's main algorithms and update process.

## G.1    $\omega$ ANALYSIS WITH SINGLE UPDATE

This section analyzes the behavior of $\omega$ for a constant $A \geq 1$ and an update $u$ positioned at the center of the sigmoid. Assuming $D$ originates from a uniform domain and is large enough to reflect this distribution, we study $\omega$ for a specific error level $\epsilon$. For an update $u$ between $k_i$ and $k_{i+1}$:

$$\arg\min \omega \; : \; \max\{M(k_{i+1}) - M'(k_{i+1}) + 1, M'(k_i) - M(k_i)\} \leq \epsilon \tag{9}$$

**Theorem 2** *For the adjustment model $M'$ and error $\epsilon > 0$, and a random update $\min D < u < \max D$, we have, $\mathbb{E}[\omega] \leq \frac{2(|D|-1)}{\max D - \min D} \ln\left(\frac{A-\epsilon}{\epsilon}\right)$.*

**Proof 2** *Let $k_i < u < k_{i+1}$, $i \in (1 : n-1)$, and define $d$ as the minimum distance from $u$ to its neighboring elements:*

$$\theta = \min\{u - k_i, k_{i+1} - u\}, k = \arg\min_{k_i, k_{i+1}} \{u - k_i, k_{i+1} - u\}.$$

*In a uniform distribution, the distance between two elements is also uniformly distributed. If $X \sim uniform(a, b)$, then for $x, x' \sim X$, the random variable $Y = |x - x'|$ follows $Y \sim uniform(0, b - a)$. Thus, the distribution of $\theta$ is $uniform(0, \max D - \min D)$. For $|D|$ elements:*

$$\mathbb{E}[quantile] = \frac{\max D - \min D}{|D| - 1}. \tag{10}$$

*Since $u$ falls into one of the $|D| - 1$ quantile:*

$$\theta \sim uniform\big(0, \mathbb{E}[quantile]\big). \tag{11}$$

*Assume $u = 0$ and $k = k_i$ (left side closest). Using the sigmoid symmetry, the same analysis applies for $k = k_{i+1}$. Then, $M'(\theta) = M(\theta) + \frac{A}{1+e^{\omega\theta}}$. From inequality 9:*

$$\frac{A}{1+e^{\omega\theta}} \leq \epsilon \rightarrow 1 + e^{\omega\theta} \geq \frac{A}{\epsilon} \rightarrow e^{\omega\theta} \geq \frac{A}{\epsilon} - 1 \rightarrow$$

$$\ln\left(\frac{A}{\epsilon} - 1\right) \geq \omega\theta \rightarrow \omega \leq \frac{1}{\theta} \ln\left(\frac{A}{\epsilon} - 1\right).$$

*Using $\mathbb{E}[\theta]$ from Eq. 11: $\mathbb{E}[\omega] \leq \frac{2}{\mathbb{E}[quantile]} \ln\left(\frac{A-\epsilon}{\epsilon}\right)$. Substituting Eq. 10 gives the result. This conclusion applies symmetrically to the right side ($k = k_{i+1}$).*

This shows the system numerically bounded and parameters change are monotone as the system receives updates.

## G.2    NEURAL NETWORK LEARNING FEASIBILITY

This section provides a theoretical framework to show that the proposed neural network operates within a feasible solution space for optimal solutions. We derive a feasibility condition and prove that a parameter configuration satisfying it always exists. Even in the worst case, where each sigmoid covers a single update, the sigma-sigmoid modifications ensure the total error near the update is limited to $\epsilon$.

The minimum distance $d$ represents the densest region, where updates affect the center with distance $d$ from it. Assuming the closest key is at the left boundary, $-d, -2d, -3d, \ldots$, the effect of $\sigma(0, A, 1, \phi)$ on the index prediction is:

$$\sum_{i=0}^{|U|-1} \frac{A}{1+e^{\omega \cdot d \cdot i}} \leq \epsilon. \tag{12}$$

Note that the size update is the same as buffer size $|U| = \rho$.

**Lemma 1** *Given Equation 12 with $\omega > 0$, $d > 0$, and $\epsilon > 0$, the upper bound for $|U|$ is:*

$$\rho = |U| \leq \frac{2}{\omega d} \ln\left(\frac{Ae^{\frac{\omega d\epsilon}{A}} - A}{\epsilon}\right). \tag{13}$$

**Proof 3** *Let $f(x) = \frac{1}{1+e^{\omega dx}}$. Using the Euler-Maclaurin formula, we approximate the sum: $\sum_{i=0}^{|U|-1} f(i) \approx \int_0^{|U|-1} f(x)dx + \frac{1}{2}[f(0) + f(|U| - 1)]$. Focusing on the integral: $\int_0^{|U|-1} f(x)dx = \frac{1}{\omega d}[\ln(1+e^{\omega d(|U|-1)}) - \ln(2)]$. Then, given $\sum_{i=0}^{|U|-1} f(i) \leq \epsilon$, we derive: $|U| \leq \frac{1}{\omega d} \ln(2e^{\omega d\epsilon} - 1) + 1 \approx \frac{2}{\omega d} \ln\left(\frac{Ae^{\frac{\omega d\epsilon}{A}} - A}{\epsilon}\right)$, so this completes the proof.*

**Theorem 3** *For any $\rho = |U| \geq 2$, $d > 0$, and $\epsilon > 0$, there exists a configuration of $\omega$ and $A$ satisfying Equation 13.*

**Proof 4** *$A = \epsilon$ simplifies the inequality $|U| \leq \frac{2}{\omega d} \ln\left(e^{\omega d} - 1\right)$. The function $f(\omega) = \frac{2}{\omega d} \ln(e^{\omega d} - 1)$ is continuous for $\omega > 0$ and diverges as $\omega \rightarrow 0^+$. Thus, for any finite $|U| \geq 2$, there exists an $\omega > 0$ satisfying the inequality.*

Theorem 3 shows that the system can achieve a parameter configuration that reduces the approximation error, regardless of the update(buffer) size. If the system fails after extensive iterations ($MaxIter$ in Algorithm 1), retraining (Section 4.2) becomes necessary, not due to the capacity of the Sigma-Sigmoid model.

## G.3    COMPLEXITY ANALYSIS

**Updates Time Complexity.** As shown in Figure 10, Sig2Model employs a multi-stage approach to handle updates while minimizing retraining frequency. The system first attempts to insert new updates into available placeholders using Gaussian Mixture Model (GMM) allocation, which has a constant time complexity of $\mathcal{O}(1)$. When no placeholders remain, updates are stored in a B+tree

buffer with logarithmic time complexity $\mathcal{O}(\log \rho)$, where $\rho$ represents the buffer's maximum capacity. Once the buffer reaches capacity ($\rho$ updates), Sig2Model performs incremental integration of the buffered updates into the model using SigmaSigmoid boosting. This operation has a time complexity of $\mathcal{O}(s + \rho \log \rho)$, where $s$ denotes the current size of the index array. Finally, when the number of active SigmaSigmoids reaches the system's capacity $\mathcal{N}$, Sig2Model initiates a full retraining of the RadixSpline model with time complexity $\mathcal{O}(N \log N)$, where $N = s + \rho$ represents the total data size (existing data plus buffered updates).

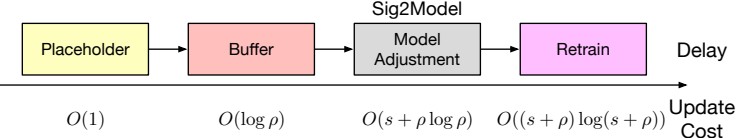

Figure 10: **Approaches to delaying retrain categorized by insertion cost.** $s$ is the size of data and $\rho$ is **buffer size.**

**Algorithms Time Complexity.** The time complexity of the algorithms in Sig2Model is as follows: (1) Lookup: $O(\text{Model} + \mathcal{N} + \log(E_\tau + \epsilon))$, as it requires inference using the learned index model, followed by $\Pi$ parameters with $\mathcal{N}$ sigmoids, and a final search of the data. (2) GMM clustering (Algorithm 2): $O(|D^0|)$, which requires one pass over the initial data. (3) Reindexing reply buffer (Algorithm 3): $O(\kappa + \rho)$, requiring a pass over both the buffer and the current reply buffer. (4) Updating the reply buffer (Algorithm 4) from $rb(t-1)$ to $rb(t)$: $O(\rho)$, requiring a single pass over the buffer size.(4) Construction Cost: The adjustment model requires a memory complexity of $O(\mathcal{N})$, if we have $NN_\Psi$ module complexity change to $O(\mathcal{N} + K)$. Additional overheads arise from tasks such as generating the initial index model, $M_0$, trained on the dataset $D^0$. These overheads depend on the primary index modeling; in our case, we used RadixSpline Kipf et al. (2020). Additionally, the process of training a neural network on $D^0$ incurs a computational complexity of $O(MaxIter \times |D^0|)$.

# H    EXPERIMENTAL SETTINGS

**Environment.** Sig2Model is implemented in C++17 and compiled with GCC 9.0.1. We use PyTorch for the neural network implementation Paszke et al. (2019). The evaluation is performed on an Ubuntu 20.04 machine with an AMD Ryzen ThreadRipper Pro 5995WX (64-core, 2.7GHz) and 256GB DDR4 RAM, and a data-centered GPU with 40GB vRAM.

**Datasets.** Sig2Model is assessed using three SOSD benchmark datasets Kipf et al. (2019): *(1) FB Dataset (2019a):* 200M Facebook user IDs, *(2) Wiki Dataset (2019b):* 190M unique integer timestamps from Wikipedia logs, *(3) Logn:* 200M values sampled from a log-normal distribution ($\mu = 0$, $\sigma = 1$). Key-value pairs are pre-sorted by key before Sig2Model initialization. All experiments use 8-byte keys from the datasets with randomly generated 8-byte values.

**Baselines.** We compare Sig2Model against: *(1) B+Tree:* A standard STX B+Tree Bingmann (2024), *(2) Alex Ding et al. (2020):* An in-place learned index, *(3) LIPP Lan et al. (2023):* A delta-buffer learned index, *(4) DILI Li et al. (2023):* A hybrid index combining in-place and delta-buffer methods. Open-source implementations are used for comparisons.

**Workloads.** QPS is measured on four workloads: *(1) Read-Only:* 100% reads, *(2) Read-Heavy:* 90% reads, 10% writes, *(3) Write-Heavy:* 50% reads, 50% writes, *(4) Write-Only:* 100% writes. Read/write scenarios interleave operations (e.g., 19 reads per write in read-heavy). The keys are randomly selected using a Zipfian distribution Barahmand & Ghandeharizadeh (2013).

**Metrics.** Evaluation metrics include: *QPS:* Average operations per second, *Latency:* 99th percentile operation latency, *Index size:* Combined size of the index and neural network model.

**System Parameters.** System parameters are tuned by sensitivity analysis: buffer size ($\rho = 1000$), replay buffer size ($\kappa = 500$), sigmoid capacity ($\mathcal{N} = 20$), RadixSpline error range (128) Kipf et al. (2020), confidence level ($\delta = 0.95$), regularization parameters ($\nu = 0.5$, $c = \gamma = 1$), $MaxIter = 100$, $\epsilon_\Pi$ and $\epsilon_\Psi$ (Algorithm 1) both set to 0.01, sampling fraction ($\lambda = 0.1$, Algorithm 4), and regularization coefficients in Equations 5 and 7 set to 1. The value $d$ is empirically determined for the initial data ($D^0$) of each dataset. $D^0$ size ($s_0$) is 50% of the respective dataset size.

# I  DETAILED EVALUATIONS RESULTS

## I.1  CPU VS. GPU TRAINING.

GPU training significantly outperforms CPU training, especially as the number of sigmoids $\mathcal{N}$ increases. Due to parallel processing, GPUs scale more efficiently, widening the performance gap at higher $\mathcal{N}$. Table 4 shows this trend—while GPU time grows moderately, CPU time increases steeply, making GPUs essential for larger model capacities.

Table 4: Training Time Comparison: CPU vs. GPU (in milliseconds)

| $\mathcal{N}$ | CPU (ms) | GPU (ms) |
|---|---|---|
| 1 | 15 | 17 |
| 5 | 241 | 28 |
| 10 | 399 | 56 |
| 20 | 955 | 108 |
| 50 | 4705 | 174 |

## I.2  GMM IMPACT ANALYSIS.

Table 5 compares $Sig2Model_{GMM}$ (GMM-based placeholder placement) with $Sig2Model_{rand}$ (random placement). $Sig2Model_{GMM}$ has 32% higher update latency and 19.5% more memory usage on average, as it strategically places placeholders based on predicted update distributions. In contrast, $Sig2Model_{rand}$ randomly places slots, leading to many unused placeholders. The lowest increase in latency and memory usage is observed for the Logn dataset due to its complex distribution.

Table 5: Normalized update latency and memory usage of $Sig2Model_{rand}$ over $Sig2Model_{GMM}$ on the write-heavy workload over different datasets.

| Dataset | Latency | Memory |
|---|---|---|
| Wiki | 43.2% | 34.6% |
| Logn | 16.3% | 4.2% |
| FB | 36.7% | 19.9% |

## I.3  SIG2MODEL PARALLELIZATION ON INFERENCE MODULE

Since the Sigma-Sigmoid model boosted by multiple weak sigmoid learners, its computations can be parallelized across multiple threads, improving performance by distributing the workload. Ideally, the number of threads can be increased to $\mathcal{N}+1$; however, there is a trade-off between the benefits of parallelization and I/O contention.

To evaluate the impact of thread parallelization, we varied the number of threads to process Sigmoid components in the Sig2Model. Results, averaged over 5 runs (Figure 11), show overhead dropping sharply to $1.0\%$ at 7 threads. Beyond this, performance degrades slightly due to result aggregation costs.

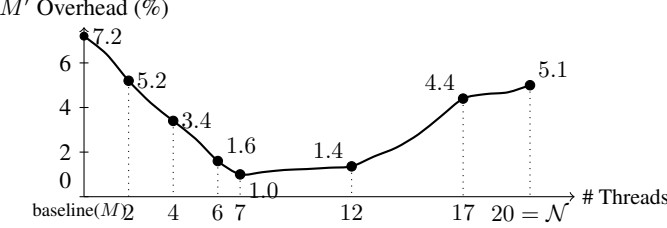

Figure 11: Overhead $M'$ compared to $M$ as a function of the number of threads. The overhead drops sharply until 7 threads, then degrades slowly due to thread coordination costs.

At zero threads means single thread for all, the overhead is $7.2\%$, decreasing rapidly with more threads. However, after 7 threads, degradation begins, reaching $5.1\%$ at 20 threads (equal to $\mathcal{N}$).

This highlights that while threading improves performance, excessive synchronization can negate its benefits.

## I.4 SENSIVITY ANALYSIS

We performed extensive sensitivity analysis on key hyperparameters. We run the experiments on the full version of Sig2Model on the Wiki Dataset with Read-heavy workload, and generalized these results for all our experiments.

Table 6: Sensivity Analysis Results

| Parameter (Tested Values) | Value 1 | Value 2 | Value 3 | Value 4 |
|---|---|---|---|---|
| Sigmoid Size $\mathcal{N}$ (5, 10, 20, 40) | 3.2 (-25.2%) | 3.9 (-8.6%) | 4.3 (opt) | 4.1 (-2.9%) |
| Buffer Size $\rho$ (250, 500, 1000, 2000) | 4.0 (-5.4%) | 4.1 (-2.6%) | 4.3 (opt) | 4.2 (-0.8%) |
| Replay Buffer $\kappa$ (250, 500, 1000, 2000) | 4.2 (-1.2%) | 4.3 (opt) | 4.2 (-0.4%) | 4.2 (-0.9%) |
| Regularization Coeff. (0, 0.5, 0.75, 1) | 4.1 (-2.9%) | 4.2 (-0.6%) | 4.2 (-0.2%) | 4.3 (opt) |
| Error Thresholds $\epsilon$ (0.1, 0.2, 0.4, 0.8) | 4.3 (opt) | 4.2 (-0.8%) | 4.1 (-2.3%) | 4.0 (-5.0%) |

**Cross-Dataset Generalizability.** While Table 6 presents results for the Wiki dataset, we observed consistent behavior across the FB and Logn datasets. The optimal $N \approx 20$ is largely dictated by the hardware's parallelization saturation point (see Appendix I.3) rather than the data distribution. Lower $N$ values ($< 10$) reduced the system's ability to buffer bursty updates common in the FB dataset, increasing retraining frequency, while higher $N$ values ($> 30$) incurred inference latency penalties that outweighed the benefits of deferred retraining across all workloads.

## I.5 TAIL LATENCY ANALYSIS

We provide comprehensive p99 latency measurements relative to median latency across different ensemble sizes ($\mathcal{N}$) using our optimal 7-thread configuration on the Wikipedia read-heavy workload.

Table 7: p99 Latency Reduction over Different $\mathcal{N}$

| Ensemble Size $\mathcal{N}$ | 5 | 10 | 20 | 40 |
|---|---|---|---|---|
| p99 Latency Reduction | -38.2% | -44.6% | -60.4% | -87.2% |

We also run the same experiment on the other baselines (Table below). Our analysis reveals that while all learned index systems exhibit some tail latency degradation due to retraining, Sig2Model demonstrates significantly better behavior (60.4% reduction vs 92-127% for baselines).

Table 8: p99 Latency Reduction over Different Baselines

| Baseline | ALEX | LIPP | DILI | Sig2Model ($\mathcal{N} = 20$) |
|---|---|---|---|---|
| p99 Latency Reduction | -92.0% | -108.1% | -127.7% | -60.4% |

This improvement stems from two key factors: (1) LIPP and DILI require frequent retraining (approximately every 500 updates), (2) While ALEX retrains less frequently, each retraining event incurs substantially higher latency. Our design achieves better tail latency by distributing the retraining overhead more evenly through the neural joint optimization framework.

**Resilience to Distribution Shifts.** Regarding dynamic workload shifts (e.g., Zipfian $\rightarrow$ Exponential), our analysis indicates that a fixed $N$ acts as a safety valve. While a dynamic-$N$ policy could theoretically absorb longer shift periods, it risks masking the need for a structural retrain, thereby degrading lookup latency with excessive sigmoid terms. Under regimes of frequent shifts, Sig2Model maintains target QPS by relying on the GMM module ($\Psi$), which adapts to the new update probability density faster than the linear model. As shown in Figure 6(c), the system effectively reduces the "recovery time" (epochs) in subsequent shifts. We assume shifts are episodic; under continuous high-magnitude variation, the system gracefully degrades to baseline retraining frequencies.

## I.6 ANALYSIS OF ENSEMBLE SIZE IMPACT ON LOOKUP

The larger ensemble sizes ($\mathcal{N}$) linearly increase inference time. We address this concern through both empirical analysis and architectural optimizations. First, our sensitivity analysis on the Wiki dataset (read-heavy workload) reveals that Query-per-second (QPS) metric improves with larger $\mathcal{N}$ up to 20, as the benefits of deferred retraining outweigh the latency costs. At $\mathcal{N} = 40$, we observe a 2.9% QPS drop (see Table 9), confirming that excessively large ensembles can negatively impact performance.

Table 9: QPS performance for different ensemble sizes $\mathcal{N}$

| Ensemble Size $\mathcal{N}$ | 5 | 10 | 20 (optimal) | 40 |
|---|---|---|---|---|
| Performance (MQPS) | 3.2 (-25.2%) | 3.9 (-8.6%) | 4.3 | 4.1 (-2.9%) |

The choice of $\mathcal{N} = 20$ represents a careful balance between update agility and lookup performance. While larger ensembles could theoretically provide greater update capacity, our experiments confirm that $\mathcal{N} = 20$ delivers optimal throughput for read-heavy workloads while still offering substantial improvements in update efficiency compared to traditional learned indexes.

Second, the parallelizable nature of the SigmaSigmoid architecture effectively compensates for the increased computation. As detailed in Appendix I.3, distributing the workload across just 7 threads reduces the parallelization overhead to a negligible 1.0% for $\mathcal{N} = 20$. This demonstrates that with proper implementation, the latency impact becomes practically insignificant for reasonable ensemble sizes.

## I.7 WORST-CASE SCENARIO ANALYSIS

Contrary to standard index structures where skewed updates often cause hotspots, Sig2Model handles narrow, dense update bursts efficiently. A concentrated burst (*e.g.*, 10k updates in a small range) manifests as a single large step in the CDF, which can be approximated by just one or two high-amplitude sigmoids ($A \gg 1$), preserving ensemble capacity.

The theoretical worst-case for Sig2Model is a *dispersed update pattern* (e.g., uniform random insertions across the entire domain). In this scenario, the CDF shifts at many distinct locations, forcing the optimization to assign a unique sigmoid center $\phi_i$ for each shift. This rapidly exhausts the fixed capacity $\mathcal{N}$ (e.g., $\mathcal{N} = 20$), triggering the fallback retraining mechanism significantly faster than clustered workloads.

## I.8 COMPARISON WITH SPECIALIZED SCENARIOS

Recent updatable indexes focus on specific workload patterns or storage constraints rather than general in-memory random updates. Liang et al. (2024) optimize strictly for sliding-window scenarios (append-heavy), and Wu & Ives (2024) target shifting workloads. Similarly, Lan et al. (2024) focus on disk-resident environments where I/O overheads dominate. Unlike these specialized approaches, Sig2Model provides robust support for general random updates in memory, making it suitable for a broader range of low-latency applications.

