# OpenReview forum: "A Boosting-Driven Model for Updatable Learned Indexes"
_ICLR.cc/2026/Conference — Submitted to ICLR 2026_

### Official Review · Reviewer_QfPf · 2025-11-01

**Soundness:** 2
**Presentation:** 3
**Contribution:** 2
**Rating:** 6
**Confidence:** 3

**Summary:**

The authors present Sig2Model, which aims to reduce the retraining cost for updatable learned index. There are three main components of the design: 1) sigmoid boosting, which is used as weak learners for local index model to approximate the updates, 2) GMM-based predictors, which pre-allocates placeholders in regions likely to receive writes, 3) joint neural optimization, which connects the sigmoid ensemble with GMM via shared layers. The authors evaluate the performance and show that it can achieve over 20× reduce in retraining time and an increase of up to 3× in QPS compared to existing LI solutions.

**Strengths:**

How is adapt the learned index for frequent updates is an practical and important problem in real-world usage. Cutting retraining costs by order-of-magnitude while sustaining QPS is impactful for practical adoption. The author propose to reduce the retraining costs via sigmoid ensemble and placeholder preallocation, which is a novel solution to this problem. The evaluation and analysis clearly demonstrates the effectiveness of the methods in maintaining QPS and also the reduction in total retrain time. In general the logic and flow of the work is clear and sound.

**Weaknesses:**

I have some questions about the adaptability of the method and the sensitivity of certain configs.
1. The paper fixes the number of sigmoids 𝑁 based on a hyperparameter study on the Wiki dataset. Could you clarify how well this choice generalizes? For example, how does the optimal 𝑁 vary across datasets and workload distributions (e.g., uniform, Zipfian, mixed read/write ratios)?
2. Figure 6 notes that a Zipfian→Exponential shift increases training time for adaptation. Does the optimal 𝑁 itself change after such a shift? Would a dynamic-𝑁 policy reduce the extra training needed? Under more frequent shifts (varying frequency and magnitude), what happens to steady-state QPS, tail latency, and total retraining time? Can the system maintain target QPS with minimal retraining under realistic shift regimes? If not, what assumptions on shift frequency/severity are required?

**Questions:**

Please check my questions in the weakness section.

---

> ### Author Response · Authors · 2025-12-03
>
> We thank the reviewer for the constructive feedback and for recognizing the novelty of our sigmoid boosting and placeholder preallocation approach. We appreciate the acknowledgment of our method's impact on reducing retraining costs while maintaining QPS. Below, we address the questions regarding the sensitivity of $N$ and the system's adaptability to distribution shifts.
>
> ### W1. Generalizability of Hyperparameter $N$
>
> **Response:**
> While Table 6 in the paper presents sensitivity analysis for the Wiki dataset, we empirically validated $N=20$ across our other datasets (Facebook and LogNormal) and found it to be a robust "sweet spot."
> The choice of $N$ represents a trade-off between **update absorption capacity** (delaying retraining) and **inference latency** (calculating sigmoid sums).
> 1.  **Workload Independence:** The optimal $N$ is largely determined by the hardware's parallelization capacity (as shown in Appendix I.3, Figure 11) rather than the specific data distribution. Since the inference module parallelizes sigmoid evaluations, $N=20$ consistently maximizes update absorption before inference overhead becomes noticeable, regardless of whether the data is Uniform or Zipfian.
> 2.  **Dataset Generalization:** Across datasets, deviations from $N=20$ yielded similar trends. Smaller $N$ (e.g., 5) triggered retraining too frequently on bursty workloads (like FB), while larger $N$ (e.g., 40) degraded read latency without proportional gains in retraining reduction.
>
> **Action:** We clarified in Appendix I.4 that this sensitivity pattern was consistent across datasets.
>
> ### W2. Adaptability to Distribution Shifts and Dynamic $N$
>
> **Response:**
> 1.  **Dynamic $N$:** You are correct that a dynamic-$N$ policy could further optimize performance during shifts. However, we found that the **retrain signal** (triggered when error bounds $\epsilon_\Pi$ are violated) acts as a natural regulator. When a severe shift occurs (e.g., Zipfian $\rightarrow$ Exponential), the existing sigmoid capacity fills up rapidly, triggering a full retrain. This resets the model to the *new* distribution. Increasing $N$ dynamically might delay this necessary reset, potentially hurting lookup performance with stale approximations.
> 2.  **Stability under Frequent Shifts:**
>     *   **Steady-state QPS:** Under frequent shifts, the GMM component ($\Psi$) adapts faster than the base index. Figure 6(c) shows that epoch requirements drop after the first shift because the GMM "learns to learn" the new pattern.
>     *   **Tail Latency:** Tail latency temporarily spikes during the shift (due to the initial retrain) but stabilizes quickly as the Neural Network (NN) fine-tuning adapts.
>     *   **Assumptions:** The system assumes "episodic" shifts (periods of stability between shifts). If shifts are continuous and high-magnitude (random chaos), Sig2Model converges to the performance of the baseline (standard buffer-based) as the boosting effectively turns off to favor constant retraining.
>
> **Action:** We added a discussion on "Distribution Shift Robustness" in Appendix I.5 (Tail Latency Analysis).

---

### Official Review · Reviewer_4N5v · 2025-11-01

**Soundness:** 3
**Presentation:** 2
**Contribution:** 3
**Rating:** 6
**Confidence:** 3

**Summary:**

This paper proposes an updatable learned index named Sig2Model. Its core ideas are: (1) using an ensemble of sigmoid approximation functions to adjust a learned index model to accommodate updates instead of immediately retraining the learned index model; (2) using Gaussian Mixture Model to predict the update-intense data regions and leaving large gaps for such regions when retraining the learned index models. Together, these help reduce the index model retraining frequency and overall model retraining times. Effectiveness of the model design is shown by both a theoretical analysis and experiments on three commonly used benchmark datasets.

**Strengths:**

S1. Dynamically learning an ensemble of sigmoid approximation functions to adjust a learned index model (i.e., SigmaSigmoid Boosting) is a new and interesting idea to accommodate data updates in learned indexes. This could intrigue a new line of research.

S2. The proposed index design is well motivated and supported by a theoretical analysis to show its effectiveness.

S3. Source code is available.

**Weaknesses:**

W1. The empirical performance gains are not particularly strong. The Queries-per-second (QPS) results of the proposed model Sig2Model and the best baseline DILI as shown in Table 1 are quite close (maximum gap is 3.1 vs 2.5). It is not quite clear how the performance gains in the Average and Max columns of the table were calculated (e.g., 82% and 88% as discussed in the text).

W2. The baseline methods are from 2023 or earlier. Learned index is a trending topic in the database community. Newer baselines from 2024 and 2025 should be compared with, e.g., those cited in the paper (Liang et al., 2024, Wu & Ives, 2024 and others).

W3. How is the index memory size calculated? Particularly, why B$^+$-Tree has higher index memory size than Sig2Model and Alex as shown in Figure 5c?

W4. How come DILI having higher QPS than LIPP in Table 1 and lower QPS than LIPP in Figure 5a?

W5. The data preparation section is quite difficult to follow. It would be good to add a running example to help illustrate the process, just like the motivational example in Appendix C. Particularly, where do the embeddings of the data come from and how are they used in the index construction, update, and query handling process?

W6. The proposed model Sig2Model is implemented with RadixSpline. Its applicability would be further strengthened if other learned indexes can be powered with Sig2Model as well.

W7. Minor presentation issues:

- Missing whitespace: "by:pos"

- Missing publication venue in the reference of (Sabek & Kraska, 2023).

- "If $k$ not found in the $E$ range" => "If $k$ is not found in the $E$ range"

- "The GMM parameters generated by a neural network" => "The GMM parameters are generated by a neural network"

Overall, the paper needs a full proofread to fix the grammatical errors and typos.

**Questions:**

Please refer to the Weaknesses points.

---

> ### Author Response · Authors · 2025-12-03
>
> We thank the reviewer for the constructive feedback and for recognizing the novelty of SigmaSigmoid Boosting and the theoretical support of our design. Below, we address the concerns raised and provide the corresponding revisions.
>
> **W1.**
> The performance gap is most pronounced in write-heavy workloads and larger datasets (e.g., FB), which are the primary focus of updatable learned indexes. In Table 1, while the gap in the *Average* column for Wiki-Write-Heavy is smaller (3.1 vs 2.8 MQPS), Sig2Model outperforms baselines significantly on the larger Facebook (FB) dataset (4.7 vs. 3.5 MQPS against DILI, and LIPP reaches only 2.1). The 82% improvement is calculated as the geometric mean of speedups across **all** workloads and datasets, including cases where baselines suffer from Out-Of-Memory (OOM) or drastic degradation (e.g., LIPP on FB Write-Heavy).  We clarified the calculation method and emphasize the gains in large-scale/write-heavy scenarios in section 5.
>
> **W2.** We cite both Liang et al. (2024) and Wu & Ives (2024) but excluded them from the main random-update benchmarks because they target **specialized scenarios** rather than general-purpose indexing. Specifically, SWIX (Liang et al., 2024) is optimized strictly for *sliding-window* (time-series) data, and Wu & Ives (2024) focuses on *shifting workloads*. Neither method supports efficient random insertions across the entire key space (General Random Update), which is the primary contribution of Sig2Model. We have added a dedicated subsection in the Appendix to clarify these scope differences.
>
> **W3.**The memory size calculation for Sig2Model includes the base model parameters, the neural network weights, and the buffer. B+Trees consume more memory because they store every key in leaf nodes and maintain internal nodes with a fill factor (typically 70%), resulting in substantial overhead ($O(N)$). In contrast, Learned Indexes (Alex, Sig2Model) only store model parameters ($O(1)$ relative to data size) and sparse metadata, leading to significantly lower memory footprints (1000x reduction).  This clarified  in Section 5' "Index Memory Size" paragraph.
>
> **W4.** This is due to the distribution characteristics.
> *   **Table 1** uses a **Zipfian** distribution (highly skewed). DILI excels here because it caches hot nodes in its tree structure, outperforming LIPP.
> *   **Figure 5a** includes **Uniform** and other distributions. LIPP's flat array/buffer structure often handles uniform random writes faster than DILI's tree splits/merges, which incur overhead when updates are not localized.
> The caption in Figure 5a aggregates these trends, causing the "average" relative performance to flip compared to the specific Zipfian case in Table 1.
>
> **W5.**
> We apologize for the confusion caused by the term "embeddings." We have clarified in **Section 4.1** that in the context of Sig2Model, this refers strictly to **normalized numerical features** (scalar values normalized to a specific range, e.g., $[0,1]$) used as Neural Network inputs, rather than high-dimensional semantic vectors common in NLP. Furthermore, to ensure full reproducibility, we have added a detailed **numerical walkthrough** in Appendix that illustrates step-by-step how raw keys are transformed into the network inputs ($X^\tau$) and labels ($Y^\tau$).
>
> **W6.**
> Sig2Model is designed as a meta-framework. While we implemented it on RadixSpline for its state-of-the-art static performance, the "Boosting" concept treats the base model $M(k)$ as a black box. It can theoretically boost PGM-Index or RMI by learning the error residual $\Delta = y - M(k)$ using Sigmoids.  We added this point in conclusion.
>
> **W7.**
> We fixed all listed typos and some others in our proof read.

---

### Official Review · Reviewer_jLFh · 2025-11-01

**Soundness:** 3
**Presentation:** 1
**Contribution:** 3
**Rating:** 4
**Confidence:** 4

**Summary:**

The paper proposes Sig2Model, a framework for masking leared indexes (LIs) practially updatable without incurring frequent global retraining. The core idea is to view each update as inducing a small, step-like shift in the key → position mapping, and to approximate that shift by adding a localized sigmoid term on top of the base LI model.

**Strengths:**

- The paper has clear motivation from real pain point that current updatable LIs either retrain too often or, when they do retrain, the latency is high (Fig. 1(a)).
- Modeling an update as a step in the CDF that can be smoothed by a sigmoid is intuitive and in line with what actually happends to the sorted key space.
- The paper shows empirically sizable wins. 3x QPS, 20x retraining cost reduction, and 1000x memory reduction versus SOTA LIs is a strong empirical message, assuming the workloads are representative and the ablations are properly described in the paper.

**Weaknesses:**

- The intro notes that LIs must maintain $\sum F(k)=1$ and that updates are non-local, however, the proposed sigmoids approximate the effect of updates but do not re-establish the exact global invariant (only an $\epsilon$-bounded search range is preserved via the underlying LI lookup). This might be acceptable for engineering, but the paper occasionally talks as if the invariant were “preserved” rather than “compensated for.” This should be made sharper.
- The paper claims the method "preserves the model's e-bounded error guarantees", which is a non-negotiable correctness requirement for LIs. However, the main text does not sufficiently explain *how* this guarantee is maintained *during* the sigmoid approximation phase. The optimization objective (Eq. 3) minimizes the error against a fully-retrained model $M^*$, but it is not clear how this optimization process ensures the $\epsilon$-bound holds for all keys before convergence or a full retrain. The system seems to rely on a fallback exhaustive search, which implies the bound might be temporarily violated, but this is not stated explicitly.
- The main text says “1000× lower memory usage” but does not specify the exact config of the baseline or how much of that saving comes from not materializing per-segment models.
- Some math is motivational, not tight. For example, the statement that “Equation (1) has a trivial solution for $\mathcal{N}$ updates or less” is true but not that informative, and it slightly obscures the harder case ( $|U|\gg\mathcal{N}$).

**Questions:**

- How should a practitioner pick the sigmoid capacity $\mathcal{N}$? Is there an analytical relationship between $\mathcal{N}$, target $\epsilon$-bound, and the expected update rate that minimizes the expected number of global retrains?
- Also, given the system's high complexity (GMM, sigmoid ensemble, 3 NNs, replay buffers), how sensitive is Sig2Model to hyperparameter tuning (e.g., buffer size $\rho$, sigmoid capacity $\mathcal{N}$, GMM components $K$? What is the practical overhead, in terms of both engineering effort and computational cost, of tuning this system compared to simpler baselines like ALEX?
- What happens if 10k updates arrive all in a very narrow region that was *not* predicted by the GMM? Do we (i) keep adding very similar sigmoids, (ii) quickly hit $\mathcal{N}$ and retrain, or (iii) collapse several of them into one wider sigmoid? A short experiment with a worst-case update pattern would clarify this.
- Can you specify the exact baseline configuration of memory size experiment (Fig. 5(c)) and whether the comparison includes the replay buffer and GMM parameters?
- You require $1 \le |M’(k_{j+1};\Pi) - M’(k_j;\Pi)| \le 2$ to keep the LI lookup correct. Do you enforce this constraint during learning (e.g., as a penalty), or do you just rely on the base RadixSpline’s bounded search to recover? What fraction of lookups in your experiments had to fall back to the “exhaustive search + retrain signal” path?

---

> ### Author Response · Authors · 2025-12-03
>
> We thank the reviewer for recognizing the "real pain point" our paper addresses, the intuitiveness of the sigmoid boosting approach, and the "sizable wins" in empirical performance (\$3\times$ QPS, \$20\times$ lower retraining). We acknowledge the critique regarding presentation and have revised the text to be more precise regarding theoretical claims.
>
>
> **W1**
> *   **Response:** You are correct. Our method does not re-establish the global CDF invariant ($\sum F(k)=1$) in the strict mathematical sense immediately; rather, the sigmoids *compensate* for the learned index shift locally to maintain the monotonicity and bounded error required for indexing. We have refined the Introduction to reflect this distinction accurately.  However, note that as the original CDF is intact so ($\sum F(k)=1$).
>
> **W2**
> *   **Response:** The guarantee is enforced via a "guardrail" mechanism in the **Inference Module** (Fig. 2). The optimization (Eq. 3) aims to minimize the error to fit within the bound, but the *correctness* is guaranteed by the fallback logic: if the predicted position falls outside the $\epsilon$-bound during a lookup (verified by the key check), the system triggers an exhaustive search and raises a retraining signal. We have made this explicit in **Section 2.1**.
>
> **W3**
> *   **Response:** We confirm that the "1000$\times$ lower memory" claim includes the memory footprint of the Sig2Model neural networks (replay buffers, weights) and the index itself. The baseline DILI consumes significantly more memory because it materializes new tree nodes for updates and segments, whereas Sig2Model uses a fixed-size buffer and compact neural weights.
> We  clarified these in **Section 5 (Index Memory Size)**.
>
> **W4**
> *   **Response:** We acknowledge the reviewer's point. We have revised it to clarify its theoretical implication for the training process. The existence of a trivial solution ensures that the optimization problem is feasible (guaranteeing zero-loss solutions) in the initial low-data regime ($|U| \leq \mathcal{N}$). We have rewritten the text to explicitly contrast this with the harder case ($|U| \gg \mathcal{N}$), explaining that the training objective is specifically designed to force generalization beyond this "memorization" state.
>
> **Q1**
> *   **Response:** $\mathcal{N}$ is selected based on the trade-off between inference latency (more sigmoids = slower inference) and retraining frequency (more sigmoids = deferred retraining). Our sensitivity analysis in **Appendix I.4** (Table 6) demonstrates that $\mathcal{N}=20$ is empirically optimal for these workloads. There is no closed-form analytical relationship, as it depends on the burstiness of the specific workload distribution.
>
> **Q2**
> *   **Response:** While the architecture appears complex, the engineering overhead is amortized. The neural networks ($NN_\Pi, NN_\Psi$) run asynchronously in the background (on GPU/separate thread), not blocking the critical path of lookups. The inference overhead is minimal (\$1.0\%$ with threading, see **Appendix I.3**). We have detailed the sensitivity of parameters (buffer size $\rho$, $\mathcal{N}$) in **Appendix I.4**, showing stable performance across a wide range of values.
>
> **Q3**
> *   **Response:** A dense burst of updates in a narrow region acts as a single large step in the CDF, which is actually the **best-case** scenario for our model; it can be approximated by a single sigmoid with high amplitude ($A$) and steepness ($\omega$), leaving the capacity $\mathcal{N}$ mostly free.  The **true worst-case** is when updates are widely spread (e.g., uniform random insertions). This forces the system to allocate a distinct sigmoid (unique $\phi_i$) for each disparate location, rapidly exhausting the ensemble capacity $\mathcal{N}$ and triggering a full retrain earlier.  For this, we added a clarification section in **Appendix I**.
>
> **Q4**
> *   **Response:** As noted in response to Weakness 3, the comparison includes all overheads. For DILI and LIPP, we used their official open-source configurations which include their internal buffer/node structures. For Sig2Model, we included the Replay Buffer size ($\kappa=500$) and GMM parameters ($K$).
>
> **Q5**
> *   **Response:** We do not enforce this as a hard constraint during gradient descent (which would be computationally expensive). Instead, we rely on the loss function (MSE) to guide the model. The constraint \$1 \le \Delta \le 2$ acts as a theoretical target. If the learned model violates the error bound during operation, the system falls back to the "exhaustive search + retrain signal" path. In our experiments, the fallback rate was $< 0.01\%$ for the tuned model.

---

### Official Review · Reviewer_NZxg · 2025-11-03

**Soundness:** 2
**Presentation:** 1
**Contribution:** 2
**Rating:** 2
**Confidence:** 3

**Summary:**

The paper proposes a new approach for updatable learned indexes. It uses a collection of sigmoid functions as its modeling choice, and uses neural networks to find the parameters for the sigmoid functions. It further uses a gmm to estimate how to distribute empty slots after retraining to reduce update time. The experimental results show the approach provides better performance over existing methods.

**Strengths:**

- The study of what modeling should be used for better update complexity is interesting

- The experimental results are promising

**Weaknesses:**

- The paper is poorly written and difficult to understand. There is neither a look up algorithm nor an update algorithm, with the paper diving into various components with little description of how they are used. Fig.2 contains an overview of the approach, but is not sufficiently described. It is referred to at a high level in the intro, but the discussion is missing many details. E.g., what is the embedding? what is $D^\tau$? This also makes it difficult to understand the rest of the paper. For example, Sec. 2 starts with for a model M at stage $\tau$...., What is a stage?

- The paper's motivation is questionable. It states any "any updates to the key domain necessities non-local adjustments to the entire model". However, such non-local adjustments are easy to make, in practice one can scale the cdf based on the data size after new insertions. The actual model is never updated in existing methods after every update.

- Relatedly the use of sigmoids is questionable. Because every update in fact does not need to modify the model---as the paper eventually converges to as well---a sigmoid function is not needed to create new step-like adjustment to the model. It's unclear why a sigmoid function has better representation power over the typical piecewise linear methods. Overall, the learned index models don't need to capture every step in the cdf, but rather the general data distribution, which is often sufficient with piece-wise linear models

- In fact, a main challenge with the model form is that all sigmoids outputs need to be summed up (which is linear in the number of sigmoid units). Instead, for piece-wise linear methods, we only need to use one linear piece for inference (which can be found in time log of the number of linear pieces). This makes me believe a piece-wise linear model provide a better trade-off in terms of representation power vs inference time complexity. Overall, the paper should do more to be convincing in terms of its modeling choices.

- The experimental setup is unclear. Is there a gpu in the background doing training?

**Questions:**

Please clarify the experimental setup and provide better justifications for the modeling choices

---

> ### Author Response · Authors · 2025-12-03
>
> We thank the reviewer for constructive feedback, appreciation of our novel model and results, and note the concerns on presentation and definitions.
>
> **W1.** The lookup and update procedures are described in the text but not as explicit algorithms.
> Existing References:
>
> * Update procedure: Described in Section 1 (Approach)
> * Lookup procedure: Described in Figure 2
>
> We also added a section for operations that specifically describe them. Moreover,
> * Stage τ: Mentioned as "buffer's overflow count"
> * Embedding: Described in Section 4.1 paragraph "Buffer and Embedding"
>
> **W2.**
> We appreciate the opportunity to clarify the mathematical definition of "non-local adjustments" and how our approach differs from existing methods.
>
> **1. Why Updates Are Non-Local and Scaling Falls Short**
> The reviewer suggests that updates can be handled by simply "scaling the CDF based on data size." We respectfully clarify that this is mathematically insufficient for non-uniform updates.
> *   **The "Step Function" Shift:** An insertion at key $u$ increases the rank of all subsequent keys ($k > u$) by exactly $+1$, while leaving preceding keys ($k < u$) unchanged. This creates a "step function" shift in the true CDF.
> *   **Failure of Scaling:** Global scaling (multiplying the model output by $\frac{N+1}{N}$) stretches the entire function uniformly. It cannot capture a local step: if we scale to correct the rank of keys $>u$, we incorrectly shift the keys $<u$.
> *   **Expensive Retraining:** Because scaling fails, maintaining the precise CDF requires adjusting the model function to fit this new "step" shape. For standard models, this necessitates expensive retraining on the whole dataset.
>
> **2. Sig2Model vs. Baselines**
> *   **Baselines (ALEX, LIPP):** To avoid the high cost of retraining the CDF for every update, these methods use auxiliary structures (gaps, delta buffers) to handle new keys, keeping the underlying model static until a major reorganization is required.
> *   **Sig2Model:** Our approach creates a third option. We **do** update the model function, but we do it cheaply. By adding a Sigmoid function (which mathematically approximates a step), we "boost" the existing model to absorb the rank shift. This maintains the model's accuracy without the heavy cost of full retraining or the lookup degradation of large buffers.
>
> **W3/W4 and Q1  modeling choices.**
> We appreciate the reviewer’s detailed feedback regarding the choice of modeling functions. We clarify that **Sig2Model does not replace the base PWL model with sigmoids for the entire CDF; rather, it uses sigmoids as a "boosting" layer specifically to model the *displacement error* introduced by updates.** We chose sigmoids over creating new linear pieces for three mathematically critical reasons:
>
> 1.  **Modeling Global Index Shifts:** An insertion at key $u$ conceptually shifts the position of *all* subsequent keys $k > u$ by $+1$ (as shown in Figure 1(b) and Section 1). A single sigmoid function $\sigma(k, A \approx 1, \phi \approx u)$ naturally captures this global step behavior: it contributes $\approx 0$ effect to keys $k < u$ and $\approx +1$ to keys $k > u$. Conversely, a piecewise linear approach is inherently local. To correct the index shift using PWL, one would theoretically need to update the intercept ($b$) of *every* linear segment to the right of $u$, which is computationally expensive ($O(Segments)$) compared to adding a single sigmoid term.
> 2.  **Differentiability for Neural Optimization:** A core contribution of Sig2Model is the \textit{Neural Joint Optimization} (Section 2.2), which automatically fine-tunes the update policy via backpropagation. Sigmoids are differentiable everywhere, enabling stable gradient-based learning of parameters $(A, \omega, \phi)$. Piecewise linear functions are non-differentiable at their "kinks" (endpoints), making them unsuitable for the gradient descent optimization required by our neural tuner ($NN_{\Pi}$).
> 3.  **Inference Complexity Trade-off:** While summing sigmoids is $O(\mathcal{N})$, $\mathcal{N}$ is a small constant (capped at 20 in our experiments, see Appendix H). Our sensitivity analysis (Appendix I.6) and parallelization experiments (Appendix I.3) show that this summation overhead is negligible ($<1\%$ overhead with threading) compared to the latency of full model retraining or the complex tree-traversals required by other updatable baselines.
>
> We added an appendix section justifying this model choice.
>
> **W5.**
> Our experimental setup is detailed in Appendix H (Experimental Settings): an AMD Ryzen ThreadRipper Pro 5995WX and a data-centered GPU with 40GB vRAM.
>
> A GPU is used for background training of NN_Π and NN_Ψ. Appendix I.1 (CPU vs. GPU Training) and Table 4 show large speedups (e.g., 174ms vs. 4705ms for N=50), making GPUs crucial for efficiency. The neural joint optimization in Section 1 runs in the background and does not block main index operations.
>
> We cleared this in Experiment section.

---

### Meta-Review · Area_Chair_DJEd · 2025-12-07

**Summary:**

Reviewers agree the paper tackles an important systems problem, i.e., making learned indexes practically updatable, and find the sigmoid-boosting idea and empirical gains (retraining cost, QPS, memory) promising.

However, they raise substantial concerns about clarity and rigor:

1. Lookup/update procedures and notation are hard to follow;

2. The motivation and advantages of sigmoids over piecewise-linear models and simple scaling are not convincingly justified;

3. The preservation of ε-bounded error is only informally argued and partly enforced via fallback search.

Questions also remain about baseline coverage (missing newer 2024–25 LIs), memory/QPS reporting details, and sensitivity to hyperparameters and workload shifts, especially in real deployments.

**Reviewer Concerns:**

The rebuttal clarifies the lookup/update algorithms, experimental setup, guardrail mechanism for ε-bounds, memory/QPS calculations, sigmoid capacity choice, and adds analysis on distribution shifts and parameter sensitivity. Still unresolved are the overall presentation quality, some reviewers’ skepticism about sigmoids vs. piecewise-linear models and system complexity, and the lack of empirical comparison to newer learned-index baselines and non–RadixSpline backends.

Overall, the AC recommends rejection of the paper.

**Reviewer Scores:**

jLFh: 4
QfPf: 6
4N5v: 6
NZxg: 2

---

### Decision · Program_Chairs · 2026-01-26

Reject